# Adding salt to expand voltage window of humid ionic liquids

Ming Chen [1,3], Jiedu Wu [2,3], Ting Ye[1], Jinyu Ye [2], Chang Zhao[1], Sheng Bi [1], Jiawei Yan [2], Bingwei Mao[2] & Guang Feng [1✉]

Humid hydrophobic ionic liquids—widely used as electrolytes—have narrowed electrochemical windows due to the involvement of water, absorbed on the electrode surface, in electrolysis. In this work, we performed molecular dynamics simulations to explore effects of adding Li salt in humid ionic liquids on the water adsorbed on the electrode surface. Results reveal that most of the water molecules are pushed away from both cathode and anode, by adding salt. The water remaining on the electrode is almost bound with $Li^+$, having significantly lowered activity. The $Li^+$-bonding and re-arrangement of the surface-adsorbed water both facilitate the inhibition of water electrolysis, and thus prevent the reduction of electrochemical windows of humid hydrophobic ionic liquids. This finding is testified by cyclic voltammetry measurements where salt-in-humid ionic liquids exhibit enlarged electrochemical windows. Our work provides the underlying mechanism and a simple but practical approach for protection of humid ionic liquids from electrochemical performance degradation.

[1] State Key Laboratory of Coal Combustion, School of Energy and Power Engineering, Huazhong University of Science and Technology (HUST), 430074 Wuhan, China. [2] State Key Laboratory of Physical Chemistry of Solid Surfaces, and Department of Chemistry, College of Chemistry and Chemical Engineering, Xiamen University, 361005 Xiamen, China. [3] These authors contributed equally: Ming Chen, Jiedu Wu. ✉email: gfeng@hust.edu.cn

Driven by the demand for the efficient use of intermittent renewable energies and electric vehicles, the development of electrochemical energy storage (EES) devices with outstanding performance has been at the forefront of energy technologies[1–4]. Nevertheless, developing safe EES devices with excellent energy density and high-power density remains a significant challenge[1–4]. One momentous component for determining the performance of the EES device is the electrolyte. Room-temperature ionic liquids (ILs), with unique properties, including excellent thermal stability, nonflammability, and especially wide electrochemical windows, are an emerging class of candidates for EES devices[5–9], such as supercapacitors[8,9], batteries[10], and solar cells[11]. However, owing to their hygroscopic nature, ILs can spontaneously adsorb water from the humid environment, regardless of their hydrophilicity or hydrophobicity[12]. The effect of water in humid ILs at the electrolyte/electrode interfaces has attracted much attention since the system performance is controlled by those electrochemical interfaces. For humid ILs, it has been found that water molecules prefer to accumulate on charged electrode surfaces[13–15]. Very unfavorable to EES devices, the surface-adsorbed water can effectively narrow the electrochemical windows of ILs due to its electrolysis, resulting in performance degradation[13,16].

Much of current research has demonstrated that the water electrosorption on the electrode is governed by the working voltage and the association of water molecules with their neighbors (including surface charges, electrode materials, and IL ions)[14,17,18]. It has been reported that due to the strong interaction with anions, water molecules are excluded from the negatively charged electrode with hydrophilic ILs[18]. However, it is still an unaddressed issue that, for hydrophobic ILs that have been widely used as electrolytes[19,20], they become humid when exposed to the atmosphere[12], and unfortunately, their water much favors adsorption on both negatively and positively charged electrodes[15,16,18]. To avoid compromising their wide electrochemical windows, it is of great importance to minimize surface-adsorbed water and its impact, and thus enhance the voltage window of humid hydrophobic ILs.

Recently, in battery community, the electrochemical window of aqueous Li-ion electrolytes is noticeably enlarged by the "water-in-salt" electrolyte[21–26]. This is ascribed to the nontrivial decrease of water activity since the strong interaction between water and Li+ leads to a large shrinkage of "free" water (i.e., water is not bound with Li+)[21–27]. Theoretical work also revealed that the water molecules could be expulsed from the positive electrode with Li+, suppressing the oxygen evolution[27,28]. Herein, analogous to the "water-in-salt" electrolyte, we investigated the effect of adding Li salt in humid hydrophobic ILs on the ion and water distributions near electrodes, using molecular dynamics (MD) simulation. The simulation system consists of a slab of humid ILs confined between two graphite electrodes (Supplementary Fig. 1). Two types of hydrophobic ILs (1-methyl-1-propylpyrrolidinium bis(trifluoromethylsulfonyl)imide, [Pyr13][TFSI], and 1-butyl-3-methylimidazolium TFSI, [Bmim][TFSI]) were taken with the salt of Li[TFSI]. Details of simulation can be referred to "Methods" and Supplementary Note 1. Our simulations found that the added salt could expand the electrochemical window of humid hydrophobic ILs. This finding was testified by our cyclic voltammetry (CV) measurements. We further delve into the molecular structure and intramolecular interaction of salt-in-humid ILs to understand the mechanism of the electrochemical window expansion, with the help of MD simulations and density functional theory (DFT) calculations, as well as infrared spectroscopy (IR).

## Results

**Ion and water distributions.** We begin our work by dissecting the electrical double-layer (EDL) structure at the electrolyte/electrode interface. Figure 1 illustrates the number density profiles of ions (Li+, [Pyr13]+ and [TFSI]−) and water molecules as a function of distance from the electrode surface with respect to the EDL potential, $\Phi_{EDL}$, which is defined as the potential across the EDL relative to the potential of zero charges (PZC). One can find that as the electrode is polarized, the counterion, [Pyr13]+ for the negative electrode and [TFSI]− for the positive electrode, moves closer to the charged surface, in consistence with previous studies[29,30], which could be understood by the increased counterion–electrode coulombic interactions[29,30]. By comparing EDL structures in humid ILs before and after adding Li salt, it can be seen that IL cations and

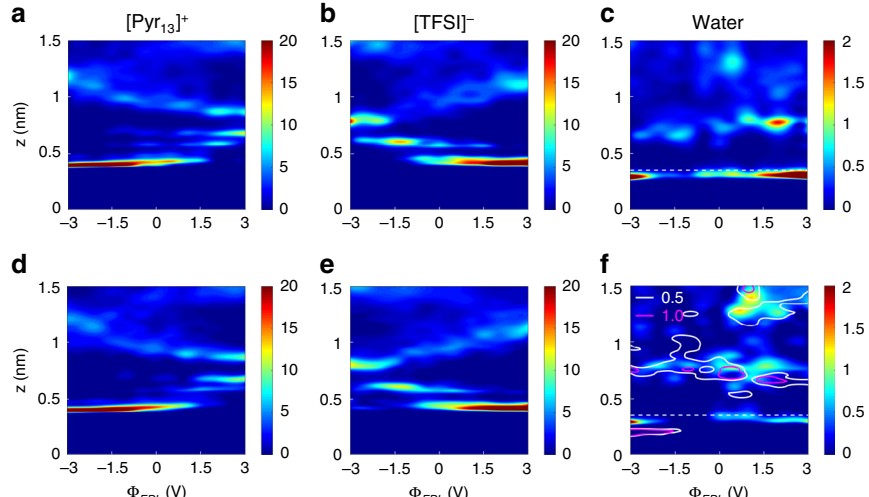

**Fig. 1 Ion and water distributions under various potentials. a–c** Number densities of cation (**a**), anion (**b**), and water (**c**) in humid 1-methyl-1-propylpyrrolidinium bis(trifluoromethylsulfonyl)imide ([Pyr13][TFSI]) as a function of distance from the electrode (z) under various potentials ($\Phi_{EDL}$). **d–f** The number densities of cation (**d**), anion (**e**), and water as well as Li+ (**f**), in salt-in-humid [Pyr13][TFSI]. The horizontal dash lines (z = 0.35 nm) in (**c**) and (**f**) represent the upper boundary of the interfacial region. The contour in (**f**) indicates the number density of Li+ ions, and the molar salt–water ratio is 1:1. Unit: # nm−3.

anions form alternating layers extending up to a few nanometers from the electrode surface, and the added salt seems to have a weak influence on the number density profiles of ILs (Supplementary Figs. 5, 6). However, the added Li salt has a quite pronounced impact on the water distribution, particularly in the interfacial region (0–0.35 nm) where water is contact-adsorbed on the electrode surface. Figure 1c and f exhibit water distributions with varying EDL potential at humid [Pyr$_{13}$][TFSI]/electrode interface and salt-in-humid [Pyr$_{13}$][TFSI]/electrode interface, respectively. These results reveal that in humid [Pyr$_{13}$][TFSI], water preferentially accumulates at the charged electrode surface, consistent with previous work of the water-in-IL mixtures[14,18,31]. In contrast, for salt-in-humid [Pyr$_{13}$][TFSI], most of the water molecules are found to be notably excluded from the electrode surface, in particular, showing a complete depletion of water under some negative polarizations and large removal of water under high polarizations. More details of the distributions of water molecules and Li$^+$ ions, as well as their evolution with working voltage, are shown in Supplementary Figs. 7–9.

Although there are still some water molecules adsorbed on the electrode surface, especially, under large polarizations, such adsorbed water molecules are found to stay closely with the added Li$^+$ ions (contour lines in Fig. 1f). We divide the adsorbed water into free and bound states, based on the number density profile of Li$^+$ ions surrounding water molecule (i.e., water is considered to be bound to Li$^+$ within a distance of around 0.25 nm; otherwise, it is labeled as free water, see Supplementary Fig. 10)[21,24,27]. We, subsequently, examine how the free and bound water in the interfacial region evolves with different polarizations and adding salt (Fig. 2a). Switching the EDL potential from 0 to −1.5 V, the electrosorbed water is nearly eliminated, and thus unable to participate in the possible electrolysis. This elimination could be ascribed to the Li$^+$–water association that could be evidenced by the similar-located peaks in the number of density distributions of

Li$^+$ ions and water molecules (Supplementary Fig. 7b). As the potential becomes more negative, the anions become fewer and further away from the electrode (see the peak height and location of anion distribution in Supplementary Fig. 5), so that Li$^+$ ions get reduced attraction from anions; meanwhile, owing to the stronger electrostatic interaction with the charged electrode, the Li$^+$ ions could overcome the energy barrier by IL ion layer[32] and then be attracted to the electrode surface (Fig. 1f). The adsorbed Li$^+$ ions are found to be associated with [TFSI]$^-$ and/or water (Supplementary Fig. 11); the adsorbed water molecules are almost bound with Li$^+$ ions, and the proportion of water in the bound state increases with the polarization (e.g., 66.38% at −2 V and 99.99% at −3 V, Supplementary Table 3), so that free water becomes depleted near the negative electrode (middle panel of Fig. 2a).

As the EDL potential increases from 0 to +1 V, there is always a layer of water molecules adsorbed on the electrode, but they are all bound with Li$^+$ ions localized near the first anion layer (~0.53 nm, Supplementary Figs. 6 and 8). Our DFT calculations show the highest occupied molecular orbital (HOMO) level of the free water is about −8.2 eV, which agrees well with previous work[33], while the HOMO level of the Li$^+$-bound water is much lower (around −15.5 eV), which is because that in the bound water the electrons from oxygen atom are donated to the Li$^+$ ion[22,34]. Thus, the lowered HOMO level by adding salt would help to enhance the oxidation stability of water on the positive electrode. It is worth noting that to quantitatively evaluate such oxidation stability, the oxidation potential could be analyzed from the free energy variation between the reactants and products under positive polarization[35–37], which has been correlated with the HOMO level: the lower HOMO leads to higher oxidation energy, indicating that more energy is required for the oxidation reaction to occur[38,39]. Therefore, the calculated HOMO level could be considered as a qualitative approach to estimating the oxidation stability of water in the electrolyte. Moreover, when the

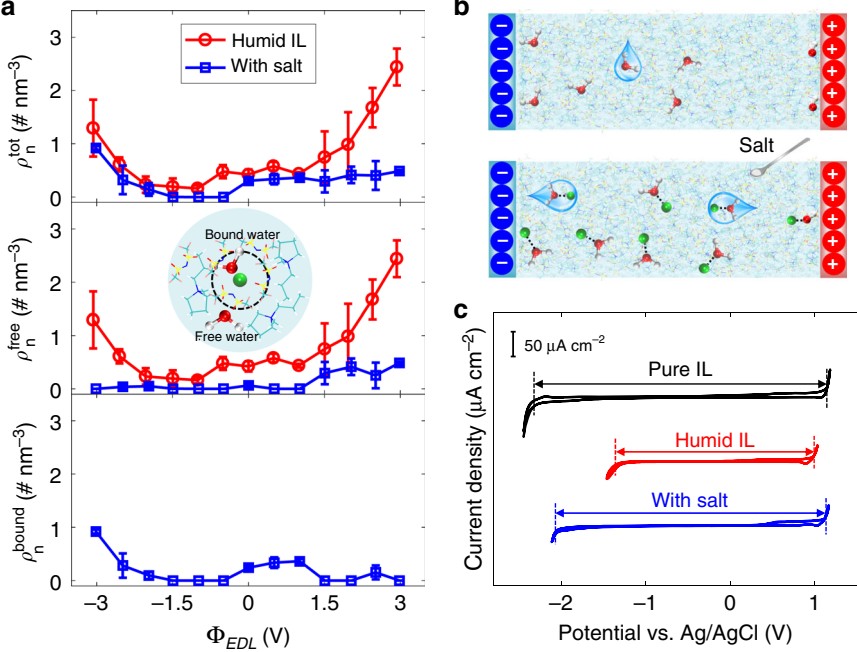

**Fig. 2 Effect of adding salt on the interfacial water and electrochemical window of electrolytes. a** Electrosorption of water from humid 1-methyl-1-propylpyrrolidinium bis(trifluoromethylsulfonyl)imide ([Pyr$_{13}$][TFSI]) with/without adding salt. The top, middle and bottom panels are, respectively, the number densities of the total ($\rho_n^{tot}$), free ($\rho_n^{free}$) and Li$^+$-bound water ($\rho_n^{bound}$) in the interfacial region. **b** Schematic of the effect of adding salt on water electrosorption. **c** Cyclic voltammograms of HOPG in pure [Pyr$_{13}$][TFSI] (black line), humid [Pyr$_{13}$][TFSI] (red line), and salt-in-humid [Pyr$_{13}$][TFSI] electrolyte (the molar salt–water ratio = 1:1, blue line). Scan rate: 100 mV s$^{-1}$. Water contents for humid [Pyr$_{13}$][TFSI] are 4392 and 4474 ppm for MD simulation and cyclic voltammetry experiments, respectively.

voltage is larger than +1 V, Li$^+$ ions are expelled from the electrode surface due to the electrostatic repulsion (Fig. 1f), and consequently, most water molecules are bound with Li$^+$ ions and then taken away from the electrode, resulting in a sharp decrease of free water (middle panel of Fig. 2a). The effect of adding Li salt on water distribution has been illustrated in Fig. 2b.

Therefore, adding salt into humid IL could not only help the water to be repulsed from both negative and positive electrode surfaces but also make the water remaining in the interfacial region become bound with Li$^+$, which could both potentially protect the humid IL from the electrolysis and thus avoid the reduction of electrochemical windows of ILs due to the existence of water. To validate this, we carried out CV measurements for pure [Pyr$_{13}$][TFSI], humid [Pyr$_{13}$][TFSI], and salt-in-humid [Pyr$_{13}$][TFSI] with highly oriented pyrolytic graphite (HOPG) electrodes. Details of measurements can be seen in "Methods" and Supplementary Note 7 (Supplementary Fig. 12). As shown in Fig. 2c, when [Pyr$_{13}$][TFSI] becomes humid (4474 ppm water, close to 4392 ppm set in MD simulation), its electrochemical window is obviously narrowed down, shrinking both cathodic and anodic voltage limits (from −2.33 ~ 1.15 V to −1.38 ~ 1.01 V). After adding salt (the molar ratio of salt to water is 1:1), the electrochemical window is clearly widened (−2.08 ~ 1.14 V), especially showing a nearly full recovery of positive polarization, compared with pure [Pyr$_{13}$][TFSI]. This phenomenon is little influenced by changing the scan rate of CV measurements (even down to 5 mV s$^{-1}$, see Supplementary Fig. 13). The simulation and experiment both demonstrate that adding Li salt into humid hydrophobic [Pyr$_{13}$][TFSI] does expand its electrochemical window. Additional modeling and experiment were carried out to find the impact of the ratio of Li salt to water, which presents the same trend for the electrosorbed water distribution and the electrochemical window (Supplementary Fig. 14).

**Origin of bound water and reinforcement of O–H bond.** In order to understand the mechanism of the voltage window expansion by adding Li salt, we first explore the underlying origin of the Li$^+$-bound water and reinforcement of O–H bond. MD simulations of bulk electrolytes were performed to analyze the stretching vibration of O–H bond of water molecules, which was used as the infrared spectroscopy (IR) reporter to monitor the structural change in humid IL electrolytes with adding salt; the spectrum of pure water was treated as the reference. The stretching vibration of O–H bond in pure water exhibits a broad IR spectrum in 3200–3500 cm$^{-1}$ (the bottom panel of Fig. 3a), corresponding to various hydrogen-bonding environments in water clusters[22,23]. Comparison of IR spectra of pure water, humid [Pyr$_{13}$][TFSI] and salt-in-humid [Pyr$_{13}$][TFSI] demonstrate a distinct modification for the characteristic of the O–H bond stretching vibration (Fig. 3a). To better delineate the change of the stretching vibration of water, three Lorentz peak functions, with each peak representing one type of water, were taken to decompose the IR spectra[40–42]. It is common and expected that the peak position of predicted stretching vibration has some deviation from the experimental result[43], and therefore, we, herein, concentrate on the shift of the peak position of the stretching vibration rather than the absolute value. Compared with pure water, IR spectra of humid [Pyr$_{13}$][TFSI] show the occurrence of new peaks at higher wavenumber (see the bottom two panels of Fig. 3a), which could be ascribed to the reduction of water clusters by the existed IL, in accord with previous work[44,45]. With adding salt, the peak positions of stretching vibrations shift towards even higher wavenumber when the salt–water ratio varies from 0:1 to 0.5:1 then to 4:1. The IR spectra of O–H stretching vibrations were measured by Fourier transform infrared spectroscopy (FTIR) for humid [Pyr$_{13}$][TFSI] with adding salt (Supplementary Fig. 15). The peak location shifts of stretching vibrations are shown in Supplementary Fig. 16. Compared with those obtained from MD simulation in Fig. 3a, it can be seen that in spite of the numerical difference, the blue-shifts (i.e., the peak location of stretching vibration shifts towards higher wavenumber) occur after adding salt, in both simulation and experiment. Moreover, IR spectra of O–H stretching vibrations were also obtained from DFT calculations for humid [Pyr$_{13}$][TFSI] (Supplementary Fig. 17), which confirm the occurrence of the blue-shift with adding Li$^+$ ion. Since all conditions are kept

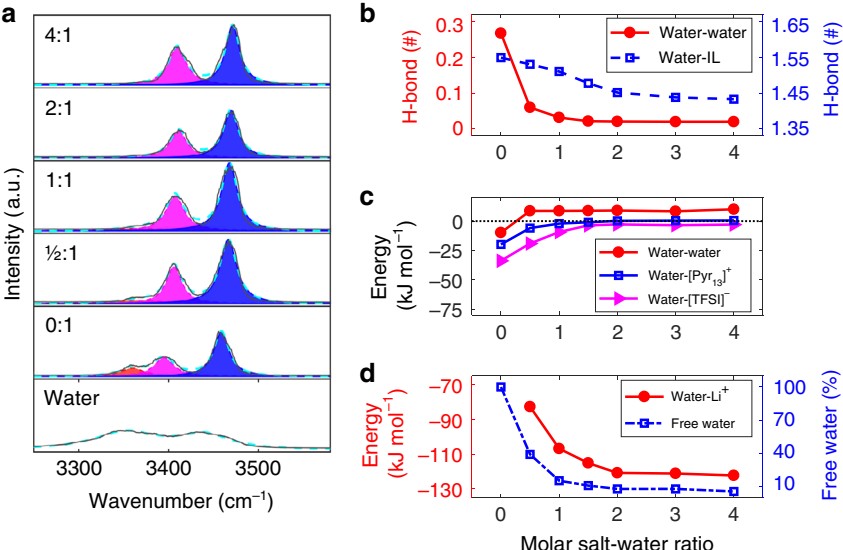

**Fig. 3 Microscopic understanding of the salt effect on water in humid ionic liquids. a** MD-calculated infrared spectroscopy (IR) spectra of O–H bond in pure water, humid 1-methyl-1-propylpyrrolidinium bis(trifluoromethylsulfonyl)imide ([Pyr$_{13}$][TFSI]), and humid [Pyr$_{13}$][TFSI] with different molar salt–water ratios. The gray lines represent IR spectra. The red, pink, and blue regions are the first, second, and third fitting spectra, respectively. The cyan dash lines represent the summation of the fitting spectra. **b** H-bonds between water molecules (left axis) and between water molecules and ionic liquid ions (right axis). **c** Interaction energy coming from van-der-Waals and coulombic interactions between water and water, [Pyr$_{13}$]$^+$ and [TFSI]$^-$. **d** Interaction energy between water and Li$^+$ (left axis), and the proportion of free water (right axis) as a function of the molar salt–water ratio.

unchanged except the amount of Li$^+$ ions, it is reasonable to ascribe the blue-shift to the Li$^+$ addition.

Essentially, the blue-shift indicates that more energy is required to cleave the O–H bond of water into radicals, and then the O–H bond is reinforced, suggesting that with adding salt, water has a lower activity and thus its electrochemical decomposition needs a higher potential[46–48]. The blue-shift is attributed to the destroyed hydrogen bond (H-bond) network[42,44]. We then computed H-bonds between water molecules themselves and between water molecules and IL ions, defined by the geometrical criterion in Supplementary Fig. 18, as a function of the molar salt–water ratio (Fig. 3b), and found that H-bonds are indeed reduced by adding salt. Especially, H-bonds between water molecules nearly disappear with the molar salt–water ratio over 1.5:1, which can also be evidenced by the dramatically diminished water clusters replaced by the Li$^+$(H$_2$O)$_n$ ($n = 1, 2, 3, 4$) complexes (Supplementary Fig. 19). Furthermore, the H-bond network in the water cluster, following a Grotthuss diffusion mechanism, helps to efficiently separate water electrolysis products[49–51], enhancing the water activity[51]. Hence, compared with humid IL, the nearly disappeared H-bond network (Fig. 3b) and largely reduced amount of water clusters (Supplementary Fig. 19b, c) could decrease the activity of water in salt-in-humid IL electrolytes.

The reduced amount of water clusters and the formed Li$^+$-bound water (Fig. 2a) could be fundamentally understood by the interaction energy between water and the other components as well as water itself (calculation of interaction energy can be seen in "Methods"). As presented in Fig. 3c, for humid ILs, the interaction between water and anions is pronounced (about $-34$ kJ mol$^{-1}$), followed by that between water and cation (ca. $-20$ kJ mol$^{-1}$) and between water and water (around $-10$ kJ mol$^{-1}$). Therefore, water molecules tend to be separated from water cluster by ions and exhibit weaker H-bond network, compared with that in pure water, which agrees with the previous work[44]. As the salt was added into humid ILs, the interaction between water and Li$^+$ rises to be the strongest, and with keeping adding Li$^+$, the interaction between water and Li$^+$ increases gradually and then gets stabilized at about

$-120$ kJ mol$^{-1}$ (Fig. 3d). Meanwhile, the coordination number of Li$^+$ ions around water increases from about 0.61 to 0.95 as the molar salt–water ratio varies from 0.5:1 to 4:1 (Supplementary Fig. 20). Very differently, the interaction between water and IL ions declines nearly to zero, and even the interaction between water molecules becomes a bit repulsive (Fig. 3c). The resultant interaction makes water molecules isolated from each other and associated with Li$^+$, leading to a sharp decrease in the proportion of free water (Fig. 3d).

Briefly, the added Li salt modifies the environment and property of water molecules on the basis of the predominated interaction between water and Li$^+$ ions, suggesting that water molecules prefer to associate with Li$^+$ ions. Consequently, the H-bond networks between water molecules and between water molecules and IL ions are disrupted, and thus the peak positions of O–H stretching vibrations of water are blue-shifted. This indicates that the strength of O–H bond is increased, and then larger energy is needed to cleave the O–H bond, suggesting that the activity of Li$^+$-bound water is lowered[46–48]. Meanwhile, the free water is largely reduced, since water tends to be bound with Li$^+$. These phenomena arising from the addition of Li salt are responsible for protecting water from electrolysis.

**Understanding the effects of adding salt on interfacial water.** We then focus on the origin of how salt affects water distributions near the electrode, by analyzing the free energy as a function of the distance to the electrode surface under different EDL potentials. The potential of mean force (PMF), which represents the variation of free energy, was evaluated by using the umbrella sampling method (see "Methods"), where the particle is allowed to rotate (Supplementary Fig. 21)[52]. For a free water molecule in [Pyr$_{13}$][TFSI] (Fig. 4a), a well-defined minimum of PMF occurs at a distance of 0.31 nm from the graphite electrode at PZC, corresponding to an accumulation of water near the electrode; with applying either negative or positive polarization, such minimum becomes more pronounced, resulting in the accumulation of water at charged electrodes. Furthermore, the free energy

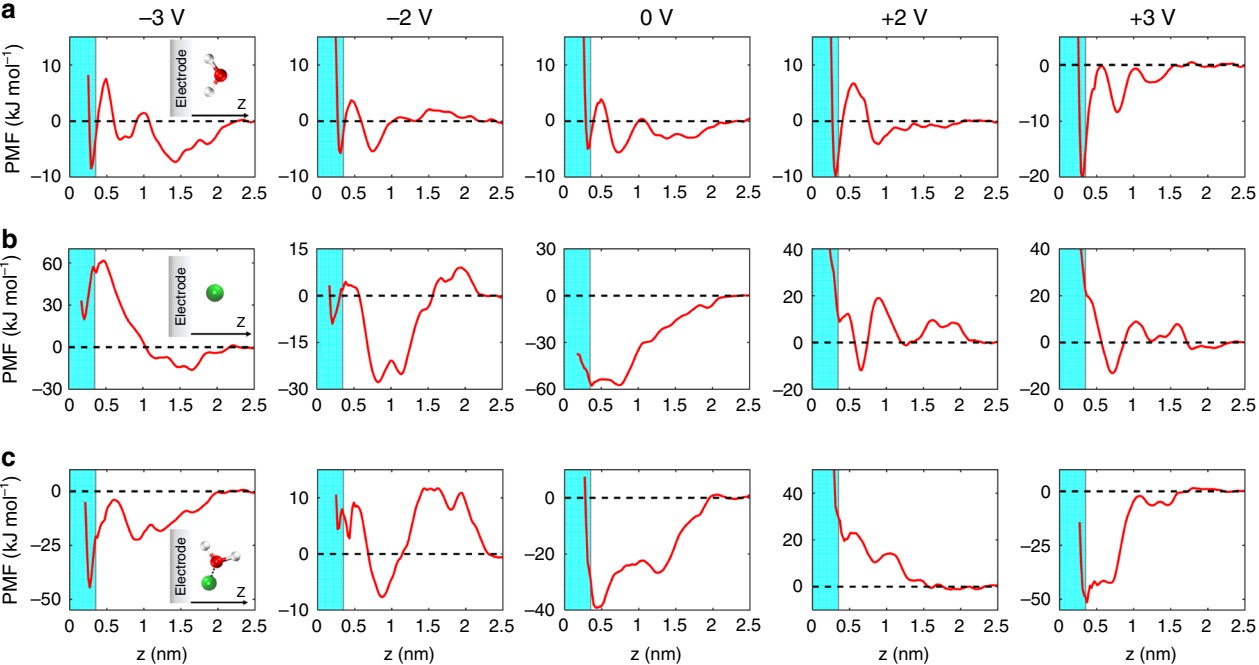

**Fig. 4 Tendency for electrosorption of water, Li$^+$ and Li$^+$-bound water at electrodes. a–c** The potential of mean force (PMF) of water (**a**), Li$^+$ (**b**), and Li$^+$-bound water (**c**) in 1-methyl-1-propylpyrrolidinium bis(trifluoromethylsulfonyl)imide ([Pyr$_{13}$][TFSI]) as a function of distance from the electrode. The cyan shaded region ($z < 0.35$ nm) is considered as the interfacial region.

of interfacial water is around −10 and −20 kJ mol$^{-1}$, respectively, at 2 V and 3 V, deeper than at −2 V (about −5 kJ mol$^{-1}$) and −3 V (around −9 kJ mol$^{-1}$), indicating that water would accumulate more at positive electrodes than at negative electrodes. However, the situation is different for Li$^+$ in [Pyr$_{13}$][TFSI] (Fig. 4b). A deep potential well appears at PZC, ranging from 0.37 to 0.74 nm, which suggests that Li$^+$ ions preferentially stay away from the electrode surface. As for negative electrode (−2 V), although a local minimum occurs at 0.21 nm from the electrode with about −9 kJ mol$^{-1}$, the lowest potential well shifts to 0.81 nm with around −28 kJ mol$^{-1}$. Under more negative polarization, the minimum of free energy curve moves further from the electrode. The phenomena that Li$^+$ ions prefer to stay out of interfacial region may be attributed to Li$^+$ ions interacting with anions that form a layer far from the negative electrode (Supplementary Figs. 5 and 7), since the intermolecular energy of ion pair, computed by interaction energies between cation and anions within its solvation shell[53], reveals that the Li$^+$-anion interaction (around −370 kJ mol$^{-1}$) is stronger than that of [Pyr$_{13}$]$^+$-[TFSI]$^-$ pair (about −220 kJ mol$^{-1}$). Under positive polarization, the potential well shows a distinct valley at around 0.71 nm in the second ion layer, due to a balance between a strong repulsion to the positive electrode and sufficient binding from the first anion layer[32], corresponding with the Li$^+$ number density profile.

Then, with adding Li salt, where could Li$^+$-bound water be? We calculated the free energy of a Li$^+$-bound water molecule, as shown in Fig. 4c. Different from the water in [Pyr$_{13}$][TFSI], the location of the first minimum of PMF at PZC shifts from 0.39 to 0.49 nm. When the EDL potential changes to −2 V, we find a positive potential well near the electrode (about 5 kJ mol$^{-1}$ at 0.27 nm), which leads to a metastable adsorption[18]. Meanwhile, the bound water induces a pronounced potential well at 0.85 nm. Accordingly, Li$^+$-bound water molecules prefer to stay far away from the electrode. On the contrary, if the EDL potential is further enlarged (either −3 V or 3 V), different from the single Li$^+$, the water-associated Li$^+$ can pass through the IL layer and become adsorbed on the electrode surface. Therefore, Li$^+$ ions tend to retard water to accumulate on the electrode surface; however, at high polarization, some water can be still dragged to absorb on the electrode surface.

The PMF curves have demonstrated where water, Li$^+$ and Li$^+$-bound water would be predominantly distributed. Specifically, the humid IL exhibits water accumulation at polarized electrodes; with adding salt, water molecules could be largely excluded from the electrode under both negative and positive polarizations. This different behavior is certainly arising from the strong association with Li$^+$ ions.

**Mechanism of expanded electrochemical window**. With above analyses, the expansion of electrochemical window could be ascribed to that the water is repulsed from the electrode surface, and the water remaining in the interfacial region becomes less active, both originating from the addition of Li salt.

To delve into the effects of adding salt on the activity of interfacial water, we evaluate the behavior of surface-adsorbed water in humid IL and in salt-in-humid IL at the electrode. The atom density profile and orientation distribution of water molecules are analyzed in Fig. 5a–d (the arrangement of interfacial water under more key voltages can be seen in Supplementary Fig. 22). The location and orientation of water molecules in the EDL are found to be sensitive to the applied voltage. Specifically, in humid ILs, as the polarization increases negatively, the interfacial water becomes ordered with the plane of water being more vertical to the electrode surface (Supplementary Fig. 22); meanwhile, under positive polarization, the interfacial water adopts a configuration nearly parallel to the electrode surface.

Then we move on to the analysis of how adding salt would affect the arrangement of water adsorbed on charged electrodes. At the negative electrode (−2 V), the interfacial water in humid ILs exhibits one sharp peak for oxygen atoms and two peaks for hydrogen atoms (left panel of Fig. 5a, b). Their dipole orientation (with peak located at ~111°, left panel of Fig. 5c) and normal orientation (peak at around 90°, left panel of Fig. 5d) illustrate that most water tends to be perpendicular to the electrode surface, with hydrogen atom pointing to the surface (left panel of Fig. 5e). Such orientations are prone to the hydrogen-evolution reaction[28]. After adding salt, the water is re-arranged: oxygen atoms of interfacial water remain in the very similar position, while the first peak of hydrogen atoms is shifted away from the electrode surface, from 0.22 to 0.25 nm (left panel of Fig. 5b); although there are still two hydrogen peaks in the interfacial region, the peak of dipole orientation of water shifts from ~111° to ~70° (left panel of Fig. 5c) and the normal orientation is nearly unchanged (left panel of Fig. 5d). These phenomena suggest that the water molecule adjusts its orientation, with its hydrogen atoms pushed away from the electrode. The change of arrangement of interfacial water under negative polarization is schematized in Fig. 5e. Meanwhile, for humid ILs at the positive electrode, the water adopts a configuration parallel to the electrode surface, evidenced by nearly the same peak location (~0.3 nm) of the oxygen and hydrogen atom density profiles and the dipole orientation peaking at 90° with the normal orientation peaking at ca. 15/165° (right panel of Fig. 5a–d); with adding Li salt, although the position of hydrogen atoms changes little, the peak location of oxygen atoms shifts away from the electrode surface (from 0.3 to 0.34 nm), and the peak of dipole orientation moves from 90° to 110°, with a small change of normal orientation (~20/160°). The re-arrangement of interfacial water under positive polarization is schematized in Fig. 5f.

Based on the arrangements of interfacial water under polarization and their changes by adding salt (Fig. 5e, f), we constructed a series of configurations with differently arranged water at graphite electrode (Supplementary Fig. 23) and performed DFT calculations to investigate how the re-arrangement could affect the charge transfer between the interfacial water and electrode (Supplementary Note 11). DFT results uncover that under negative polarization, the arrangement change due to adding salt will allow a few electrons to transfer from the electrode to the interfacial water (Supplementary Fig. 24a), making it more difficult for the hydrogen-evolution reaction. Meanwhile, under positive polarization, the re-arranged water tends to lose a few electrons to the electrode (Supplementary Fig. 24b), facilitating the inhibition of water oxidation. Moreover, MD-obtained O–H spectra in the interfacial region are also found to vary with adding salt (Fig. 5g), that is, the peak positions of O–H stretching vibrations of interfacial water in salt-in-humid IL are shifted towards higher wavenumber, indicating that O–H bond becomes more stable than that without salt[46–48].

We further testify the generality of the conclusion that the adding salt expands the electrochemical window of humid hydrophobic ILs, by combined simulation-experiment work of another hydrophobic IL [Bmim][TFSI]. Although the cations [Pyr$_{13}$]$^+$ and [Bmim]$^+$ are different in terms of their molecular structure, shape/size, and weight, observations in simulation and experiment follow the same trend (Supplementary Figs. 25 and 26), where water molecules stay away from the electrodes owing to the presence of Li$^+$ and the electrochemical window is enhanced (with increases of 0.51 V and 0.21 V, respectively, for negative and positive polarization), compared with humid [Bmim][TFSI].

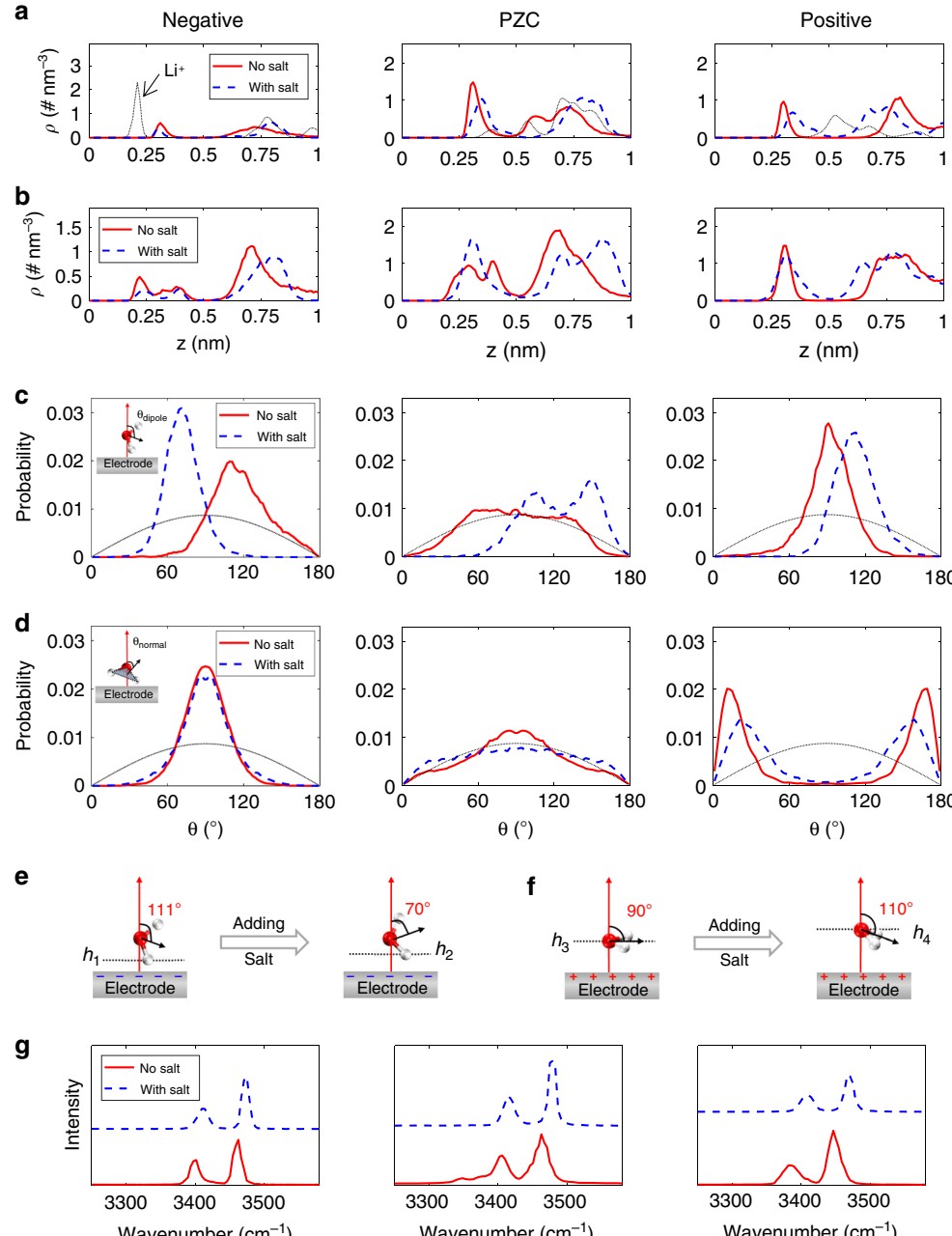

**Fig. 5 Effect of adding salt on the structure of interfacial water in humid ionic liquids. a** The oxygen atom number densities ($\rho$) of water in humid ionic liquid (IL) (red solid line), and in salt-in-humid IL (blue dash line) at electrodes. The black dotted line is the atom number density of Li$^+$ in salt-in-humid IL. **b** The hydrogen atom number densities of water in humid IL and in salt-in-humid IL at electrodes. **c, d** The dipole (**c**) and normal (**d**) orientations of interfacial water in humid IL (red solid lines) and in salt-in-humid IL (blue dash lines) at electrodes. $\theta_{dipole}$ is defined as the angle between the normal of the electrode surface and the water vector, and $\theta_{normal}$ is the angle formed between the normal of the electrode surface and the normal of water plane. The black dotted line represents the orientation of bulk water. **e, f** Schematics of arrangement change of interfacial water under negative (**e**) and positive (**f**) polarizations in humid IL due to adding salt. **g** Infrared spectroscopy (IR) spectra of interfacial water in humid IL and in salt-in-humid IL. The electrical double-layer (EDL) potentials are −2 V and 1 V for negative and positive electrodes, respectively.

## Discussion

We have investigated the effect of adding salt in humid hydrophobic ILs on distributions of ion and water at the electrode surface. Combining MD simulations, DFT calculations and CV measurements, we found that the electrochemical stability window of salt-in-humid ILs is indeed expanded, compared with the humid one. MD simulations and DFT calculations provide a molecular understanding of how and why adding salt could dramatically enhance the electrochemical window of humid ILs.

Such enhancement is ascribed to water repulsed from the electrode surface and the lowered activity of water remaining in the interfacial region, by adding Li salt.

The water is found to have a stronger association with Li$^+$ than IL ions. Therefore, since Li$^+$ ions mostly prefer to stay away from the electrode surface, water tends to be pulled away from the electrode under positive and moderately negative polarizations, extremely reducing the water in the interfacial region. For water molecules still adsorbed on the electrode surface, most of them

are bound with $Li^+$, resulting in significantly decreased activity, since the strength of O–H bond in bound water is increased due to the association with $Li^+$, and thus more energy is needed to split the O–H bond[48]. Meanwhile, the added $Li^+$ modifies the arrangement of adsorbed water by changing its orientation and atom position. Explicitly, the arrangement change of interfacial water under negative polarization leads to a reduction of electrons transferred from the electrode to water, making it difficult for the hydrogen-evolution reaction to occur; the oxygen atom is pulled away from the positively charged electrode, protecting water from electrolysis. The lower HOMO level of bound water is a complementary mechanism for the thermodynamic manner to improve electrolyte oxidation stability under positive polarization[22,34].

Although the above understanding is revealed by classical MD simulation joint with DFT calculations, and could qualitatively explain the expansion of electrochemical window, the molecular simulation with reactive force fields, which could directly mimic the processes of chemical bond breaking/forming, may give a quantitative prediction of electrochemical window expansion. Moreover, it should be noted that the approach of adding salt is tested in two hydrophobic ILs herein, and taking into account the cost of lithium salt, more ILs and cheap salts[54,55] are preferred in further investigation.

The findings reported with the concept of salt-in-humid ILs could extend the comprehension of the preferential species electrosorption, and provide a guideline for minimizing water adsorption and altering the structure and property of adsorbed water. The mitigation of adsorbed water and its impact could improve the practical performance of EES systems as an electrochemical window of humid hydrophobic ILs can be enlarged, and may also benefit other applications such as IL gating[56], lubrication[57], and electrowetting[58]. The understanding of adding salt to expand the electrochemical window, including the decrease of interfacial water, reinforcement of O–H bond, and water rearrangement, as well as the lowered HOMO level of bound water, may be responsible for the working mechanism of water-in-salt electrolytes[21–26] in battery and supercapacitor.

## Methods

**MD simulation**. MD simulations were utilized to investigate the effect of added salt on the ion and water distributions in the EDL of humid IL. Specifically, we employed MD simulation of hydrophobic IL-water-salt mixtures confined between two atomically flat graphite electrode surfaces as shown in Supplementary Fig. 1. All-atom model was taken for two hydrophobic ILs ([Pyr$_{13}$][TFSI] and [Bmim] [TFSI])[59,60]; the SPC/E model was adopted for water[61]. The OPLS force field was used for $Li^+$ ion[62]; carbon atoms in the electrode were modeled using the force fields in ref. [63]. The sizes of all simulation systems were chosen as long enough to reproduce the bulk-like state of electrolytes in the central region between two electrodes. The number of the species and the distance between two electrodes are given in Supplementary Table 1 for all systems.

All simulations were performed in NVT ensemble with MD package GROMACS[64]. The temperature was controlled through the Nosé–Hoover thermostat[65,66] at 333 K with coupling constant of 1.0 ps. A cutoff distance of 1.2 nm was employed for the van-der-Waals term via direct summation. The long-range electrostatic interactions were computed via PME method[67]. An FFT grid spacing of 0.1 nm and cubic interpolation were used to compute the electrostatic interaction in the reciprocal space. A cutoff length of 1.2 nm was adopted in the calculation of electrostatic interactions in the real space. The leapfrog integration algorithm was taken to solve the equations of motion, with a time step of 2 fs. Specifically, in order to ensure an adequate description of the electrode polarization effects in the presence of electrolytes, constant potential method[2,68,69] was implemented to allow the fluctuations of the charges on electrode atoms. To guarantee the accuracy, the electrode charges were updated on the fly every simulation step. For each simulation, the MD system was first heated at 700 K for 3 ns and then annealed to 333 K over a period of 2 ns, following by another 10 ns to reach equilibrium. After that, a 20-ns production was performed for analysis. Each case was repeated three times with different initial configurations to certify the accuracy of the simulation results.

**PMF calculation**. PMF represents the variation of free energy and is evaluated by the umbrella sampling method[70]. A series of initial configurations are generated, and each of them corresponds to a location where the molecule is restrained via a harmonic potential. It could be understood as the molecule is bound by a spring and moves back and forward around a given position, traveling freely along the plane parallel to the electrode surface. Rather than being fixed, the molecule is allowed to rotate, experiencing all possible orientations. With the weighted histogram analysis method[71], the PMF is subsequently extracted. To determine the PMF profile for water, $Li^+$ and $Li^+$–$H_2O$, one water molecule, $Li^+$–$TFSI^-$ pair, and $Li^+(H_2O)$–$TFSI^-$ pair are, respectively, added into the ionic liquid system, since the PMF for one particle (i.e., an atom, molecule, ion or group) is enough to reveal its free energy distribution and the origin of its density distribution[70,72].

**Interaction energy calculation**. We characterized the interaction energy between components A and B based on MD-obtained trajectories. Technologically, the A–B interaction energy, coming from van-der-Waals and coulombic interactions, was calculated between component A and component B surrounding A, using a cutoff method (1.2 nm). Such analysis has been used in previous modeling work[73,74].

**Quantum molecular orbital analysis**. DFT calculations on the molecular orbital of water and water/$Li^+$ complexes were performed with the Gaussian 09 program (revision, D.01)[75]. The basis sets implemented in the Gaussian program were used. The geometries of water and water/$Li^+$ complexes were fully optimized using B3LYP/DFT method with 6–311 G** basis set. Based on their optimized geometries, the HOMO energy levels were calculated at the same level.

**Experimental materials and measurements**. ILs [Pyr$_{13}$][TFSI] and [Bmim] [TFSI] were purchased from IoLiTec in the highest available quality (99%); salt LiTFSI (99.95%) was purchased from Sigma-Aldrich. Prior to each measurement. ILs were purified through ultrapure water (Milli-Q, 18.2 MΩ·cm) and then vacuum-dried for 24 h at 80 °C in a glovebox filled with ultrapure Argon (Linde Industrial Gases, 99.999%) to remove water as much as possible, and then used as dry ILs. The capacity of pure ILs [Pyr$_{13}$][TFSI] and [Bmim][TFSI] to absorb water from environment was examined under constant humidity (43.72 ± 2.45% humidity for [Pyr$_{13}$][TFSI] and 43.00 ± 1.68% for [Bmim][TFSI], see Supplementary Fig. 2). Humid ILs were prepared via adding ultrapure water into ILs. Salt-in-humid ILs were prepared by adding ultrapure water and LiTFSI into ILs, and then the mixtures were stirred up for 12 h till homogeneous solutions were formed. Detailed information of electrolyte preparation can be seen in Supplementary Note 2 with Supplementary Table 2. Even when the molar salt–water ratio reaches 4:1 (the highest used in this work), the solubility limit of LiTFSI in such two humid ILs still does not occur in the experiment (Supplementary Fig. 3)[76]. Water contents were determined by Karl Fischer Coulometer (Metrohm, KF-831)[77]. Cyclic voltammetry measurements were carried out in a glovebox by using an Autolab electrochemical workstation (Eco Chemie, The Netherlands). HOPG was used as working electrode. A piece of tape was pressed onto the flat surface and then pulled off to form a newly cleaved clean surface for electrochemical measurements. A scanning tunneling microscope image for HOPG (Supplementary Fig. 4), conducted on a scanning probe microscope (Bruker Corp., multimode 8), reveals that the proportion of carbon atom number at the step edge is estimated to be $\sim 2.1 \times 10^{-4}$, indicating that the effect of step edges is negligible in this work. Silver wire and platinum wire were used as reference electrode and counter electrode, respectively. The potentials of Ag wire in all electrolytes were calibrated with respect to an Ag/AgCl reference electrode. The time dependence of the open-circuit potential (OCP) of Ag quasi-reference electrode versus Ag/AgCl electrode in different solutions was measured (Supplementary Fig. 12), and such OCPs for different electrolytes were listed in Supplementary Table 4. The electrochemical cell used for measurements was a sealed one, which was isolated from the environment, and CV measurements were typically completed within half an hour, so that the change of water contents can be negligible. FTIR measurements were conducted on a Nexus 8700 spectrometer (Nicolet) equipped with a liquid-nitrogen-cooled MCT-A detector. A commercial demountable liquid cell (Harrick Scientific Corporation) was used. The sample was prepared in a glovebox and transferred to a spectrometer with a sealed vessel. The sample preparation for IR measurements was completed within two minutes. Then, an unpolarized IR radiation sequentially passed through two CaF$_2$ windows with a thin-layer solution (25 μm). The IR transmission spectrum of the sample was taken from 1111 to 4000 cm$^{-1}$ with a resolution of 4 cm$^{-1}$ and averaged 100 times.

## Data availability

The data that support the findings of this study are available from the corresponding author upon request. Source data are provided with this paper.

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

## Acknowledgements

G.F., M.C., T.Y., C.Z., and S.B. acknowledge the funding support from the National Natural Science Foundation of China (51876072 and 51836003). J.W.Y., B.W.M., J.D.W., and J.Y.Y. acknowledge the support of Natural Science Foundation of China (21673193, 22072123, and 21727807). G.F., M.C., T.Y., and C.Z. also thank the Hubei Provincial Natural Science Foundation of China (2020CFA093). The computation is completed using Tianhe II supercomputer in National Supercomputing Center in Guangzhou.

## Author contributions

G.F. conceived this research. G.F. and J.W.Y. designed the work of simulation and experiment, respectively. M.C., T.Y., C.Z., and S.B. carried out all simulations; M.C. did all DFT calculations. J.D.W., J.Y.Y., J.W.Y., and B.W.M. carried out the experiment. M.C. drafted the paper. All authors contributed to the analysis and discussion of the data and revision of the paper.

## Competing interests

The authors declare no competing interests.
