## [Peer Review File · Nature Communications]

REVIEWER COMMENTS

Reviewer #1 (Remarks to the Author):

Review of Chen et al.

March 23, 2020

The manuscript by Chen *et. al.* discusses the influence of the presence of lithium salt in a water-rich aprotic ionic liquid system on the onset of the hydrogen evolution/oxygen evolution reactions, i.e. the electrochemical window. In this paper, the authors use MD simulations to study the spatial distribution of species (water and ions) at the interface at different potentials. The authors compare their results to experimental data, specifically cyclic voltametry and IR spectroscopy. The major finding is that water is strongly bound to the lithium ion, which results in a higher over potential for the hydrogen evolution, while the oxygen evolution shows minor effects. This is somewhat surprising as it should be the oxygen evolution that is affected even more so (where the electrode is positively charged), if this argument is correct (or at least if it is the major driver for this behaviour). Nonetheless the anodic peaks appear to be affected less. In contrast, the major conclusion that water is strongly bound to lithium is not surprising at all, as such I see much improvement for this work. Yet the topic is highly interesting and warrants publication in a major outlet. The simulation work itself seems well done, but given a number of questions as follows, I cannot recommend publication at this point. Major revisions are required as follows:

Major concerns

- My first major concern is that the authors use a thermodynamic argument to conclude on a kinetic barrier increase - i.e. the increased over potential of the water decomposition. The validity of this approach should be discussed, in particular given that the authors use standard force fields, and no reactive force fields. Water splitting will involve hydrogen adsorption to step edges of the HOPG in this setup, it is completely unclear at this point whether the simulation derived conclusions are related to the restrictions of the simulation, or whether the thermodynamic argument is sufficient to explain the kinetic difference. Please elaborate on this.
- My second major concern is that the authors need to include discussions of orientation effects in particular in figure 4. Here the PMF is calculated by fixing the molecule/ pair of interest in terms of an umbrella sampling. I find that this approach completely ignores the orientation of water or the ion/water pair, as such panel a/c offer a very restricted picture of the

actual physics. I can see that the authors looked into orientation in Figure 5, but not in terms of the PMF. Please discuss and justify.

- It will be important to see a plot of the angular distribution of water at key potentials, as this may play an important role, in particular at the cathodic side. How is water oriented with respect to the electrode surface? How does this arrangement influence the barrier, i.e. the over potential? This will in my mind be the most important insight from such a work, but may require more simulations.
- General setup: The authors should explain in detail how close the simulation and experimental result are comparable in terms of the water concentration. In simulations they use 42 Li/water pairs, on 420 IL cation/anion pairs. There is no information on the experimental ppm once Lithium is added, a quick estimate gives about 1/4 mol of water/L at 4500 ppm, which means 1/4 mol of LiTFSI is needed in experiments to reach 1:1. This is past, or at least approaching, solubility limits of LiTFSI in some ILs. It is unclear here - please add more info so that this can be assessed properly (update e.g. table 1 in SI).
- Lines 92-93: The authors show that the affinity of Li⁺ ions for adsorption on the negatively charged electrode surface is higher than for ionic liquid cations. Please consider explaining this phenomenon in more detail. What is the proportion of Li⁺ coordinated to water molecules and Li⁺ remaining in association with the TFSI anions? Can this be better quantified? Also, how is the water arranged in this situation.
- Given the discussion so far, water is dragged to the negative interface with the lithium, but it does not react, because the Li⁺ binds it strongly. Vice versa is true for the positive polarisation, yet the positive polarisation shows less impact. Why? As the devil's advocate I will argue that the water may easily split into a proton and a hydroxide (remaining with the lithium), so in the end, if the authors would use reactive force fields, the reaction may even become more efficient at negative potentials, due to water coordination with lithium at the interface. Please comment on this, and how a reactive force field may alter the results.
- Line 126: The correlation of measured and simulated IR data is completely unclear. Figure 10 in the SI shows very little ratio change, and it is unclear what the black solid line refers to. A ratio plot for all peaks will be much better to understand the data and how it compares. Please provide such a plot.
- The discussion of the IR data is also quite confusing and not in line with the plots. Please consider rewriting. E.g. please describe qualitatively, why the coordination of Li⁺ to water is preferred over the coordination of Li⁺ to IL anions? Also include a list of peak shifts and relative comparison of peak shifts of experiment and theory.

- The authors show that Li⁺ ions preferentially stay away from the electrode surfaces but in lines 92-93 the authors say that the Li⁺ ions are attracted to the electrode surface. Please clarify. This also applies to discussion in lines 247-249. The location of Li⁺ ions as well as their coordination to water molecules should be discussed with a proper scrutiny and orderly throughout the paper.
- Line 239 Discussion: Have similar effects of salt addition on the expansion of electrochemical windows in IL systems been discussed in previous works? The discussion does not currently include a proper 'state-of-the-art' related to this topic. Also the work may benefit from expanding the discussion section by including some more detailed insight from literature. Please consider also enriching the discussion by referring to the experimental results from similar systems.
- In many places, the discussion of the results is rather qualitative than quantitative. I mean using expressions such as: more interaction, more or less bound, etc. Can the authors review their discussion of the results in more quantitative manner using coordination numbers, bond energies? Can the authors also quantitatively discuss the occupancy of species near the charged surfaces in terms of ratios of different species Li:water:IL?
- Please consider to add a 'Conclusions' section
- A general concern: Please clarify what "bound water" is by definition in the different situations described in the manuscript.
- In Figure 2c and were-else appropriate, please correlate the silver reference potential to standard Ag/AgCl or any other standard reference potential (also please mention these parameters in experimental section). I also doubt the choice of this electrode. It is not very stable in the solutions used in this work. Please provide an OCP of this electrode in the different solutions, and also correct the CVs accordingly. What is the error of this electrode in terms of the reference potential error? This is important.

Minor concerns and typos

- Figure 1: It will be nice to flip x/y axis for all panels, to enhance comparison with CV data (see also Figure 2 where this is the case).
- Figure 1: Please discuss why ions are generally further away around the potential of zero charge.
- Figure 2b is very misleading, where firstly F is not defined in the figure's caption. I assume it is a force. Presumably, the authors were trying to mark F as the repulsion force to indicate that the Li⁺ cations drive the water molecules away from the surface during anodic polarization shown on the right hand side (red). However, I don't understand why the Li⁺-associated water molecule is still being driven away from the surface during

the cathodic polarization in the exactly same way shown on left hand side (blue). The direction of the repulsive force and the schematic orientation of the molecule does not make sense in the second case, please clarify this.

- throughout the paper: Naming 'cation' is misleading in the text and in all the figures because it can refer either to Li+ or to ionic liquid cation. Please specify this everywhere precisely.
- Methods section: Experimental method sections have not enough details about preparation of solutions for experimental IR measurement.
- Line 11: lowered the activity of what?
- Line 74: distributions - typo
- Line 465 (Figure 1): At a first glance, the figure labelling 'cation' may be misleading as both Li+ and [Pyr13]+ are cations; Please specify that the label 'cation' in subplots a) and d) refers only to [Pyr13]+. It could be also beneficial for the figure readability to highlight better that the distribution of Li+ is only shown in the subplot f).
- Line 472 (Figure 2): Please explain why in humid IL (with no Li+ added) there is more water electro sorbed in positive polarization than in a negative polarization.
- Line 472 (Figure 2): subplot a) symbol of number density ρ on y axis is not explained in the figure caption
- Line 92: Typo: more 'negative' not 'negatively'
- Line 95: '*more water molecules are driven into the interfacial region, but they are almost bound with Li+...*' What do the authors mean by saying that the water molecules are 'almost' bound with Li+? Please clarify by using a more specific expression to describe the nature of bonding between Li+ and water molecules
- Line 472 (Figure 2): Subplot a) bottom panel. Please clarify in the figure label that the bound water refers to water bound to Li+ cations.
- Line 109: Typo: water remaining not remained
- Line 110: '*make the water remained in the interfacial region become bound...*' For better readability, please clarify the expression 'bound' here; for example water coordinated to Li+ cations
- Line 480 (Figure 3): Please explain in the Figure caption how data shown in subplots b), c), d) was obtained.
- Line 134: The text describes that the spectra of water in Figure 3a (bottom panel) corresponds to 'pure water', but the figure label says it is 'spectra of water in humid IL'. Please clarify.

- Line 149: '*The reduced water clusters*' or the reduced amount of water clusters, please rephrase for clarity, also the bound water should be clarified as "Li+ bound water".
- Line 149-168: '*interaction between water and water*' Please use more specific terms when quantifying interaction energy between water molecules, cations, and ILs throughout this paragraph. Do authors mean energy of hydrogen bonding? Please clarify. Same goes for Figure 3 caption (lines 481-489).
- Line 163 '*water molecules prefer to tangle with Li+ ions*' please consider replacing 'tangle' with more scientific vocabulary - coordinate etc.
- Line 166 '*indicating the reinforced of O-H*' What do authors mean by this? Please explain in more detail when O-H gets reinforced and why this suggest lowered water activity?
- Line 495 Figure 4: there is no green shaded region there
- Line 487 Figure 3c: What is meant by cation here? Li+ cation or ionic liquid cation [Pyr13]+? Please specify this throughout the text when it might be confusing.
- Line 230 '*makes O-H more stable*' What does the stability of O-H mean here? Please clarify
- Line 10: The water remained on the electrode is almost bound - The water (which) remained
- Line 193: find a positive potential well near electrode find a positive potential well near (the) electrode

Reviewer #2 (Remarks to the Author):

This manuscript reports a method for expanding the voltage window of humid ionic liquids. The idea, which is very simple, consists in adding a Li ion-based salt. The lithium "traps" the water molecules far from the interface, resulting in a substantial broadening of the electrochemical window. Experiments were performed to validate the idea and simulations allowed to understand the mechanisms at play. This work is very well conducted; it inspires on the series of recent works concerning water-in-salt electrolytes and more generally on studies where water acts as a reactant rather than as a solvent. I am sure it will interest a broad range of scientist and I recommend publication of the manuscript in Nature Communications, provided that the following points are addressed by the authors:

1/ I think that the experiments could be consolidated to really prove the importance of the effect. In fact, it is possible that the proposed mechanism affects only the kinetics of the water reduction and/or oxidation, so it would be necessary to show cyclic voltammeteries at slower rate in order to prove that nothing happens.

2/ I am quite confused by the discussion of Figure 4 (i.e. page 7). In fact the authors do not say to which system(s) (i.e. concentration) the results correspond to, so it is difficult to understand. In addition, there is a reference to Fig. 3c (line 191) which should in fact be Fig. 4c I think

3/ How was the Li-anion binding energy (line 185) calculated?

4/ I think the authors should be a bit more cautious when discussing the interaction energies (page 6). These are effective energies, computed using a classical force field and for a given structure of the liquid, and not absolute energies. The variations with Li-salt concentrations are interesting, but the relative values between the different terms should not be overinterpreted. In addition the method used to compute these terms should be detailed.

5/ There are quite a lot of English mistakes that should be corrected:

-line 74: "distributions"

-line 84: "to closely stay" does not seem right

-line 92: "negative"

-line 109: "remain"

-line 126: I do not understand "deployed" in this context

-line 173: "by using the"

-line 183: "could be attributed"

-line 205: Sentence should be rephrased ("attributed to that" does not seem right)

-line 270: do the authors really mean "adapted"? Or "adopted"?

We have studied all the comments carefully and made point-to-point response to each comment. Specifically, reviewer's comments are copied in blue, and each comment is followed by our response in black. The manuscript has been revised, accordingly, in red. After each response to comment, a brief summary is provided of what has been changed or added and where they are positioned in the revised manuscript and corresponding Supplementary Information.

To carefully make response to all the comments, we have accomplished a series of supplementary but necessary modeling and experimental work, performed the corresponding analysis, and reflected it all in the new, majorly revised version of the manuscript. Due to the heavy workload of additional modeling and revision, we added one author who has been involved in the MD simulation and analysis.

For your information, we briefly summarized the work done for this revision:

- 1) DFT and MD simulations were newly performed and analyzed to provide more evidences and detailed discussions on the expansion of electrochemical window from the thermodynamic/kinetic view.
- 2) Results and analyses of the orientation of interfacial water under key potentials were added based on performed MD simulation, and the arrangement effect of interfacial water on the charge transfer between graphite and water was explored by DFT calculations.
- 3) Experiment was conducted to obtain the time-dependence of the open-circuit potential (OCP) of Ag quasi-reference electrode vs. Ag/AgCl reference electrode in different solutions. Accordingly, all the CV curves have been updated based on the obtained OCP.
- 4) Cyclic voltammetry measurements were carried out, at different scan rates (5, 10, 50 and 100 mV s⁻¹), for HOPG in pure RTILs, humid RTILs and salt-in-humid RTILs electrolytes to examine the influence of scan rate on the electrochemical window.

- 5) Method and consistent discussions about interaction energies were provided.
- 6) Other detailed information about modeling and experiment methods was provided:
(1) water/salt concentration, (2) preparation of solutions, (3) characterization of HOPG surface, and (4) calculations of PMF and interaction energy.
- 7) Other new figures/data and discussions were added: (1) association of IL and water with interfacial Li^+ , (2) definition of H-bond and (3) discussions and outlook with referring to some literature.

Responses to Reviewer #1's Comments

Reviewer #1 (Remarks to the Author):

The manuscript by Chen et. al. discusses the influence of the presence of lithium salt in a water-rich aprotic ionic liquid system on the onset of the hydrogen evolution/oxygen evolution reactions, i.e. the electrochemical window. In this paper, the authors use MD simulations to study the spatial distribution of species (water and ions) at the interface at different potentials. The authors compare their results to experimental data, specifically cyclic voltametry and IR spectroscopy. The major finding is that water is strongly bound to the lithium ion, which results in a higher over potential for the hydrogen evolution, while the oxygen evolution shows minor effects. This is somewhat surprising as it should be the oxygen evolution that is affected even more so (where the electrode is positively charged), if this argument is correct (or at least if it is the major driver for this behaviour). Nonetheless the anodic peaks appear to be affected less. In contrast, the major conclusion that water is strongly bound to lithium is not surprising at all, as such I see much improvement for this work. Yet the topic is highly interesting and warrants publication in a major outlet. The simulation work itself seems well done, but given a number of questions as follows, I cannot recommend publication at this point. Major revisions are required as follows:

Response:

We thank the reviewer very much for carefully assessing our work and giving lots of quite constructive comments/suggestions directing more work and discussions needed to clarify our conclusions.

Major concerns

Comment 1) My first major concern is that the authors use a thermodynamic argument to conclude on a kinetic barrier increase - i.e. the increased over potential of the water decomposition. The validity of this approach should be discussed, in particular given that the authors use standard force fields, and no reactive force fields. Water splitting will involve hydrogen adsorption to step edges of the HOPG in this setup, it is completely unclear at this point whether the simulation derived conclusions are related to the restrictions of the simulation, or whether the thermodynamic argument is sufficient to explain the kinetic difference. Please elaborate on this.

Response:

We appreciate the reviewer for her/his very useful comments on the thermodynamic argument and the compatibility of simulation setup with the HOPG.

Here we would like to first address the reviewer’s comment that the influence from the possible “step edges of the HOPG”, since it is critical for modeling setup. It is true that the step edges are more active than the basal plane of HOPG. However, in our experiment, we used HOPG with very high quality (i.e., the step edges are very few), which would be evidenced by our new data from scanning tunneling microscopy (STM) measurements. To detect the possible defect of HOPG, we conducted STM on a scanning probe microscope (Bruker Corp., Multimode 8). An obtained STM image is shown in **Figure R1.1**, which exhibits that only one step edge was observed at a scan area of $0.5 \times 0.5 \mu\text{m}^2$. The proportion of atom number at the step edge is estimated as $\sim 2.1 \times 10^{-4}$, indicating that the effect of step edges could be negligible for HOPG used in this work. Therefore, in our simulation systems (both MD and DFT) we omit the step edges of the graphite electrode, by modeling the atomically flat electrode as graphene sheets that are in periodic boundary condition along the plane of electrode surface (**Supplementary Figure 1a**).

Figure R1.1 | STM image of HOPG electrode surface.

Then, we focus on the reviewer’s the comment on the thermodynamic argument. In published work of water-in-salt electrolytes, some of researchers have tried to make a correlation between certain thermodynamic/kinetic factors and expanded electrochemical windows of water-in-salt electrolytes.¹⁻⁵ However, on the fundamental level, there is still unclear about how the individual thermodynamic and

kinetic factors would specifically contribute to the electrochemical windows expansion.⁵ In particular, before this work, there is no published study showing that adding salt could expand the electrochemical windows of humid RTIL system, and its underlying mechanism.

In this work, adding salt to expand the electrochemical window could be understood by four factors in thermodynamic/kinetic perspectives: (1) *decrease of interfacial water*. Since most Li^+ ions prefer to stay away from electrode surface, water, strongly associated with Li^+ , would be pulled away from the electrode, thus dramatically reducing the interfacial water; (2) *arrangement change of interfacial water*. For water molecules adsorbed on the electrode surface, most of them are bound with Li^+ , and the added Li^+ modifies the arrangement of interfacial water, so that the HER and OER become more difficult; (3) *reinforcement of O-H bond in bound water*. The association with Li^+ reinforces O-H bond in bound water, which leads to decreased activity of bound water; (4) *lowered HOMO level of bound water*. The association with Li^+ could make the electrons of oxygen atom in bound water shift towards Li^+ ion, so that OER occurs under a higher potential. From a thermodynamic or kinetic perspective, these factors may be classified as: factors 1-2 in kinetic perspective and factors 3-4 in thermodynamic perspective.

We also fully agree that the above four factors explaining the expansion of electrochemical window can be understood (*probably quantitatively*) by the reactive force fields. However, these factors could be qualitatively proved by DFT calculations and MD simulations with classical force fields. In particular, we add new data/figures and corresponding analyses/discussions to solidify factors 1-2, and extend discussions to strength factor 3. Below, we will show them one by one.

For factor 1 (*decrease of interfacial water*): as shown in **Fig. 1f** and **Fig. 2a** of the main text in the revised manuscript, interfacial water molecules were found to decrease after adding salt. As presented in **Fig. 3c-d** of the main text, the interaction between water and Li^+ is much stronger than that between water and $[\text{Pyr}_{13}]^+$ or

[TFSI]⁻, suggesting that water prefers to be bound with Li⁺. PMF of Li⁺ in **Fig. 4b** reveals that Li⁺ ions preferentially stay far away from electrode surface, thus, bound with Li⁺, water would be pulled away from the electrode, leading to the decrease of interfacial water (i.e., the reduction of interfacial water concentration).

For factor 2 (*arrangement change of interfacial water*): for the water molecules still adsorbed on the electrode, most of them are found to be bound with Li⁺. As shown in **Fig. 5a-d** of the main text, the arrangement of interfacial water is modified by adding Li⁺. Explicitly, at negative electrode (-2 V), most interfacial water in humid IL shows one O-H bond pointing to the electrode surface and one hydrogen atom adsorbed on the surface (dipole orientation peaked at ~111° and normal orientation peaked at 90°, named arrangement-1, see **Figure R1.2a**). After adding salt, the arrangement of interfacial water changes to a configuration with the peak of dipole orientation (~70°) and similar normal orientation (named arrangement-2, see **Figure R1.2b**), and the first peak of hydrogen atoms is pushed away from the electrode surface (i.e., from 0.22 to 0.25 nm). Under positive polarization (+1 V), the interfacial water, adopting a configuration parallel to the electrode surface (dipole orientation peak at ~90° and the normal orientation peaks at 15° and 165°, named arrangement-3, see **Figure R1.2c**), changes to a configuration with dipole orientation peak at 110° and normal orientation peaks at 20° and 160° (named arrangement-4, see **Figure R1.2d**), after adding salt.

Figure R1.2 | Effect of adding salt on the interfacial water arrangement of humid RTIL at the electrified surface. a-b, The schematics of the arrangement of interfacial water under negative polarization (-2 V) in humid RTIL, named arrangement-1 (**a**) and adding salt, named arrangement-2 (**b**). **c-d**, The schematics of the arrangement of interfacial water under positive polarization (1 V) in humid RTIL, named arrangement-3 (**c**) and after adding salt, named arrangement-4 (**d**).

To understand how the water arrangement and its change affect water electrolysis (i.e., the charge transfer between interfacial water and the electrode), four typical water arrangements were taken from MD simulations (i.e., four arrangements shown in **Figure R1.2**) to construct configurations for DFT calculations. The detailed setup of DFT calculations can be found, in response to the comment on water orientation in **Major Comment 3**. Results from DFT calculations reveal that due to adding salt, the water arrangement changed from **Figure R1.2a** to **Figure R1.2b** leads to a reduction of electrons transferred from graphite electrode to water, making it difficult for the hydrogen evolution reaction taking place under negative polarization (**Figure R1.3a**). As water arrangement changes from **Figure R1.2c** to **Figure R1.2d**, the water molecule tends to lose fewer electrons to the electrode (**Figure R1.3b**), facilitating the inhibition of water oxidation under positive polarization.

Figure R1.3 | The net charge on water molecule with different arrangement near polarized electrode. a, Under negative polarization, the arrangement change (from arrangement-1 to arrangement-2) leads to a reduction of electrons transferred from electrode to water. **b,** Under positive polarization, with adding salt the water molecule tends to lose few electrons to the electrode.

For factor 3 (*reinforcement of O-H bond in bound water*): as shown in **Fig. 3** in the main text, with adding salt, the peak of O-H stretching vibration of water shifts to higher wavenumber, when the molar salt-water ratio varies from 0 to 0.5 then to 4. The blue-shift (i.e., the peak location of stretching vibration shifts towards higher wavenumber) of the O-H stretching vibration indicates that more energy is needed to

cleave the O-H bond into radicals,^{6,7} and thus, the O-H bond strength is increased, which could lead to the decreased activity of water and the increase of the electrochemical window.¹⁻³ Detailed explanation can be seen in response to **Minor Comment 23**.

For factor 4 (*lowered HOMO level of bound water at positive electrode*): based on our previous DFT calculations, the HOMO level of free water is about -8.2 eV, while that of the Li⁺-bound water is about -15.5 eV, which is because the association with Li⁺ could make the electrons of oxygen atom in bound water shift towards Li⁺ ion, raising its oxidation potential.¹

Brief summary:

The above discussion has been incorporated into the revised manuscript (*Conclusion & Discussion Section* of the main text, *Pages 11-12, lines 309-318*; *Conclusion & Discussion Section* of the main text, *Page 12, lines 323-327*; *Experimental materials and measurements Section* of the main text, *Page 15, lines 408-412*). More detailed descriptions for the STM image of the HOPG electrode surface are provided in the updated **Supplementary Note 3 (HOPG surface characterization)** with **Supplementary Figure 4 (Page 7, lines 92-103)**.

Specifically, the modification in main text is listed as:

“The water is found to **have stronger association with Li⁺ than IL ions**...water is **intended** to be pulled away from the electrode under **positive and moderately negative** polarizations ...**and thus more energy is needed to split the O-H bond**⁴³. Meanwhile, the added Li⁺ modifies the arrangement of adsorbed water, **by changing its orientation and atom position**. Explicitly, **the arrangement change of interfacial water under negative polarization leads to a reduction of electrons transferred from the electrode to water**” (*Conclusion & Discussion Section* of the main text, *Pages 11-12, lines 309-318*)

“**Although the above understanding is revealed by classical MD simulation joint with DFT calculations, and could qualitatively explain the expansion of**

electrochemical window, the molecular simulation with reactive force fields, which could directly mimic the processes of chemical bond breaking/forming, may give a quantitative prediction of electrochemical window expansion.” (*Conclusion & Discussion Section* of the main text, *Page 12, lines 323-327*)

“A scanning tunneling microscope image for HOPG (**Supplementary Figure 4**), conducted on a scanning probe microscope (Bruker Corp., multimode 8), reveals that the proportion of carbon atom number at the step edge is estimated to be $\sim 2.1 \times 10^{-4}$, indicating that the effect of step edges is negligible in this work.” (*Experimental materials and measurements Section* of the main text, *Page 15, lines 408-412*).

Comment 2) My second major concern is that the authors need to include discussions of orientation effects in particular in figure 4. Here the PMF is calculated by fixing the molecule/ pair of interest in terms of an umbrella sampling. I find that this approach completely ignores the orientation of water or the ion/water pair, as such panel a/c offer a very restricted picture of the actual physics. I can see that the authors looked into orientation in Figure 5, but not in terms of the PMF. Please discuss and justify.

Response:

We thank the reviewer for this useful comment on the calculation of PMF. In our previous manuscript, we lacked a clear description on the method of calculating PMF, although orientation effect of water has been taken into account, when using the umbrella sampling method, that is, the water molecule or pair of interest is allowed to rotate, rather than being fixed.

Specifically, when using umbrella sampling method to calculate PMF, a series of initial configurations are generated, and each corresponds to a location where the molecule or pair of interest is restrained via a harmonic potential.⁶ Taking water as an example, it could be understood as the water was bound by a spring: water could move back and forward around a given position along the direction vertical to the electrode surface, and travel freely in the plane parallel to the electrode surface. For instance, 48 different configurations of water, shown in **Figure R1.4a**, were constructed. With the weighted histogram analysis method,⁶ the PMF curve (**Figure**

R1.4b) was obtained by performing MD simulations with 48 different configurations. In simulation, the water molecule, rather than being fixed, is allowed to rotate, experiencing all possible orientations.⁷ **Figure R1.4d-g** show the orientations of water molecule at several typical sampling points (0.29, 0.47, 0.75, and 2.35 nm). It can be seen that the orientation distributions at these four sampling points are quite different from each other. In particular, the orientation of water molecule at ~ 0.29 nm shows the water is much more ordered than that at ~ 2.35 nm, and the later exhibits a random angular distribution, nearly same with that in the bulk humid ionic liquids, since the water is far from the electrode surface.

Figure R1.4 | The PMF information of water at -2 V. a, Weights of weighted histogram analysis method with 48 configurations of MD simulations. **b**, The extracted PMF based on the weighted histogram analysis method.⁶ **c**, The schematic of orientation distribution of water molecule. θ_{dipole} is defined as the angle between the normal of electrode surface and the water vector, and θ_{normal} is the angle formed between the normal of electrode surface and the normal of water plane. **d-g**, The orientation distribution of water molecule at corresponding sampling window. Red and blue lines in **d-g** represent the dipole and normal orientation. The black dash lines in **d-g** represent the orientation of bulk water.

Brief summary:

Based on the reviewer's advice, we added a clear description of PMF calculation in the revised manuscript (*Page 8, lines 204-206; Method Section* of the main text, *Page 13, lines 367-373*). More details for the orientation of water in different sampling points using the umbrella sampling method have been incorporated into the revised **Supplementary Information**, as **Supplementary Note 10** (*Pages 23-24, lines 306-333*) with **Supplementary Figure 21**.

Specifically, the modification in main text is listed as:

“... evaluated by using the umbrella sampling method (see **Methods**), where the particle is allowed to rotate (**Supplementary Figure 21**).⁴⁴” (*Page 8, lines 204-206*)

“PMF represents the variation of free energy and is evaluated by using the umbrella sampling method.⁶² A series of initial configurations are generated, and each configuration corresponds to a location where the molecule is restrained via a harmonic potential. It could be understood as the molecule is bound by a spring and moves back and forward around a given position, traveling freely along the plane parallel to the electrode surface. Rather than being fixed, the molecule is allowed to rotate, experiencing all possible orientations.” (*Method Section* of the main text, *Page 13, lines 367-373*)

New reference on the method of PMF is added, as:

“62.Allen, T. W., Andersen, O. S. & Roux, B. Molecular Dynamics — Potential of Mean Force Calculations as a Tool for Understanding Ion Permeation and Selectivity in Narrow Channels. *Biophysical Chemistry* 124, 251-267 (2006).”

Comment 3) It will be important to see a plot of the angular distribution of water at key potentials, as this may play an important role, in particular at the cathodic side. How is water oriented with respect to the electrode surface? How does this arrangement influence the barrier, i.e. the over potential? This will in my mind be the most important insight form such a work, but may require more simulations.

Response:

We appreciate the reviewer for these thoughtful comments.

Following the reviewer's suggestion, we plot angular distributions of water at several key potentials (-2.5, -2, -1, -0.5, 0, 0.5, 1, 2 V) with analyzing five more potentials (**Figure R1.5**). Under negative polarizations, interfacial water in humid IL adopts a configuration with one O-H bond pointing to the electrode, since the first peak of hydrogen is closer to the surface than that of oxygen (**Figure R1.5a1-b1**); both dipole and normal orientations of interfacial water change from random distribution to ordered ones (**Figure R1.5c1-d1**). With adding salt, since water is depleted at -0.5 and -1 V, there is no orientation distribution of interfacial water; for -2 V and -2.5 V, the dipole orientation peak shifts towards smaller angle (**Figure R1.5c1**), while the normal orientation of interfacial water changes little (**Figure R1.5d1**), suggesting that the water is tilted to make hydrogen atom staying further away from the surface (**Figure R1.2b**). Under positive polarization, in humid RTIL, the water is nearly parallel to the electrode surface: the second peak for hydrogen gradually disappears and hydrogen and oxygen have the similar peak location (**Figure R1.5a2-b2**). These indicate that the interfacial water becomes more ordered with increasing the positive polarization (**Figure R1.5c2-d2**). After adding salt, the interfacial water at 0.5 V and 1 V is re-arranged to make oxygen staying further away from electrode surface due to the association of Li^+ (**Figure R1.5a2-b2**); while at 2 V, no water is associated with Li^+ since Li^+ is pushed away from electrode surface, and thus the interfacial water adopts very similar arrangement in humid RTIL and salt-in-humid RTIL.

To understand how the water arrangement and its affect water electrolysis (i.e., the charge transfer between interfacial water and the graphite electrode), four typical water arrangements were taken from MD simulations (under -2 V and +1 V, before and after adding salt) to construct a series of configurations for DFT calculations. As shown in **Figure R1.2**, these four water arrangements are (1) water in humid RTIL at -2 V (arrangement-1, **Figure R1.2a**), (2) water in salt-in-humid RTIL at -2 V (arrangement-2, **Figure R1.2b**), (3) water in humid RTIL at 1 V (arrangement-3, **Figure R1.2c**), and (4) water in salt-in-humid RTIL at 1 V (arrangement-4, **Figure R1.2d**), with detailed data listed in **Table R1.1**.

Figure R1.5 | Effect of adding salt on the interfacial water of humid RTIL at the electrified surface. a-b, The oxygen (a1-a2) and hydrogen (b1-b2) number densities of water in humid RTIL (red solid line) and in salt-in-humid RTIL (blue dash line) at electrodes. The black dotted line in a1-a2 is the atom number density of Li^+ in salt-in-humid RTIL. **c-d,** The dipole orientation (c1-c2) and normal orientation (d1-d2) of interfacial water in humid RTIL (red solid lines) and in salt-in-humid RTIL (blue dash lines) at electrodes. The black dotted line in c1, c2, d1 and d2 represents the orientation of bulk water.

Table R1.1 | The orientations and the position of oxygen and hydrogen atoms of water molecule with respect to the graphite surface.

	negative electrode (-2 V)		positive electrode (1 V)	
	no adding salt	adding salt	no adding salt	adding salt
	arrangement-1	arrangement-2	arrangement-3	arrangement-4
θ_{dipole}	111°	70°	90°	110°
θ_{normal}	90°	90°	15°/165°	20°/160°
O	0.310 nm	0.300 nm	0.300 nm	0.340 nm
H1	0.217 nm	0.244 nm	0.280 nm	0.314 nm
H2	0.358 nm	0.390 nm	0.320 nm	0.340 nm

With water in typical water arrangements taken from MD simulations (i.e., four arrangements shown in **Table R1.1**), our new DFT systems have the graphite electrode represented by a 6×8 graphene periodic supercell containing 48 carbon atoms with 20 Å vacuum. As shown in **Figure R1.6**, for each water arrangement, we consider *four different adsorption sites* as general setup for water adsorbed on the graphite electrode.⁸ That is, the water is on top of two adjacent carbon atoms (sites T1 and T2), the center of a carbon-carbon bond (site B), and the hexagon center (site C). For arrangement-1 and arrangement-2, the hydrogen atom, closer to the surface, is on top of the adsorption site; for arrangement-3 and arrangement-4, the oxygen atom is on top of the adsorption site. DFT calculations with differently-arranged water on the graphite electrode were performed for different adsorption sites, by Vienna ab-initio simulation package (VASP).⁹ Perdew-Burke-Ernzerhof (PBE) exchange-correlation functions of generalized gradient approximation (GGA) were employed in DFT calculations.¹⁰ The projector augmented wave (PAW)¹¹ method with a cutoff energy of 400 eV was used to describe the interaction between nuclei and electrons. The convergence of energy was employed as 10⁻⁴ eV. The dipole correction and spin polarization were added for the calculations. The Γ -centered k-point meshes of 3×3×1 were adopted. Additionally, for a correct treatment of physisorption interaction,¹² the Grimme's D3 dispersion correction⁸ was employed in DFT calculations.

Figure R1.6 | Arrangement of water molecule adsorbed on graphite. **a-b**, The structures of water molecule adsorbed on graphite on four different adsorption sites with arrangement-1 (**a**) and arrangement-2 (**b**). The hydrogen atom of water, closer to graphite, is on top of four different adsorption sites, i.e. carbon atoms (sites T1 and T2), the center of a carbon-carbon bond (site B), and hexagon center (site C). **c-d**, The structures of water molecule adsorbed on graphite on four different adsorption sites with arrangement-3 (**c**) and arrangement-4 (**d**). The oxygen atom of water is on top of four different adsorption sites, i.e. carbon atoms (sites T1 and T2), the center of a carbon-carbon bond (site B), and hexagon center (site C).

Based on the DDEC6 charge analysis,¹³⁻¹⁵ the charge transfer between water

molecule and graphite is evaluated. As seen in **Figure R1.3a**, the arrangement change (from arrangement-1 to arrangement-2) leads to a reduction of electrons transferred from electrode to water, making it difficult for the hydrogen evolution reaction taking place under negative polarization (**Figure R1.3a**). Meanwhile, compared with arrangement-3, water with arrangement-4 has fewer positive charges, indicating that adding salt will make water molecule more likely to lose fewer electrons to the electrode (**Figure R1.3b**), facilitating the inhibition of water oxidation under positive polarization.

Brief summary:

Discussions on 1) the arrangement (atom density and orientation) of water at more key potentials and 2) the influence of the arrangement of water on the charge transfer between interfacial water and the electrode have been integrated into the revised manuscript (*Pages 9-10, lines 249-287*). More details on the atom density distribution and orientation of water at more key potentials are provided in the updated **Supplementary Note 11** with **Supplementary Figure 22** (*Pages 25-26, lines 334-362*), and the DFT calculations to determine how the water arrangements affect water electrolysis have been updated in **Supplementary Note 11** with **Supplementary Figures 23-24** (*Pages 26-29, lines 363-408*).

Specifically, the modification in main text is listed as:

“The arrangement of interfacial water under more key voltages can be seen in **Supplementary Figure 22** ... Specifically, in humid RTIL, as the polarization increases negatively, the interfacial water becomes ordered with the plane of water being more vertical to the electrode surface (**Supplementary Figure 22**); meanwhile, under positive polarization, the interfacial water adopts a configuration nearly parallel to the surface.

Then we move on the analysis of how adding salt would affect the arrangement of water adsorbed on charged electrodes. At negative electrode (-2 V) ... for hydrogen atoms (left panel of **Fig. 5a-b**). Their dipole orientation (with peak located at 111 °,

left panel of **Fig. 5c**) and normal orientation (peak at 90° , left panel of **Fig. 5d**) ... pointing to the surface (left panel of **Fig. 5e**) ... the water is re-arranged: oxygen atoms of interfacial water keep in the very similar position, while ... from 0.22 to 0.25 nm (left panel of **Fig. 5b**) ... the peak of dipole orientation of water shifts from 111° to 70° (left panel of **Fig. 5c**) and the normal orientation is nearly unchanged (left panel of **Fig. 5d**) ... The change of arrangement of interfacial water under negative polarization is schematized in **Fig. 5e** ... the dipole orientation peaking at 90° with the normal orientation peaking at $15/165^\circ$ (right panel of **Fig. 5a-d**) ... and the peak of dipole orientation is moved from 90° to 110° , with very small change of normal orientation ($20/160^\circ$). The re-arrangement of interfacial water under positive polarization is schematized in **Fig. 5f**.

Based on the arrangements of interfacial water under polarization and their changes by adding salt in **Fig 5e-f**, we constructed a series of configurations with different arranged water at graphite electrode (**Supplementary Figure 23**), and performed DFT calculations to investigate how the re-arrangement would affect the charge transfer between the interfacial water and electrode (**Supplementary Note 11**). DFT results show that under negative polarization, the arrangement change due to adding salt will allow a few electrons to transfer from the electrode to the interfacial water (**Supplementary Figure 24a**), making it more difficult for the hydrogen evolution reaction. Meanwhile, under positive polarization, the re-arranged water tends to lose a few electrons to the electrode (**Supplementary Figure 24b**), facilitating the inhibition of water oxidation.” (Pages 9-10, lines 249-287)

Comment 4) General setup: The authors should explain in detail how close the simulation and experimental result are comparable in terms of the water concentration. In simulations they use 42 Li/water pairs, on 420 IL cation/anion pairs. There is no information on the experimental ppm once Lithium is added, a quick estimate gives about 1/4 mol of water/L at 4500 ppm, which means 1/4 mol of LiTFSI is needed in experiments to reach 1:1. This is past, or at least approaching, solubility limits of LiTFSI in some ILs. It is unclear here - please add more info so that this can be assessed properly (update e.g. table 1 in SI).

Response:

We appreciate the reviewer for constructive comments. Following the suggestion, we listed water concentrations of all MD systems in **Table R1.2** and experimental data of the water content for humid IL and salt-in-humid IL electrolytes in **Table R1.3**.

Table R1.2. Setup parameters of simulation. Number of cation, anion, water molecules, and Li^+ , the molar salt-water ratio and the distance, L, between the electrodes, as well as the water content in each molecular dynamics (MD) system. Cation in the table means the cation in ionic liquid, i.e., $[\text{Pyr}_{13}]^+$ and $[\text{Bmim}]^+$, respectively. Anion in the table means the anion ($[\text{TFSI}]^-$) in ionic liquid.

		system	cation	anion	water	Li	salt-water ratio	L (nm)	water content (ppm)
[Pyr ₁₃] [TFSI]	Humid IL	System1	420	420	42	0	0	10.0	4392
	Salt-in-humid IL	System2	420	462	42	42	1:1	10.6	4105
		System3	420	504	42	84	2:1	11.1	3853
[Bmim] [TFSI]	Humid IL	System4	430	430	52	0	0	10.0	5168
	Salt-in-humid IL	System5	430	430	52	52	1:1	10.8	4775

Note: the water-IL ratio in humid IL is equal to that in salt-in-humid IL, and the smaller water concentration in salt-in-humid IL is because of the increase in total mass of electrolyte with adding salt.

Table R1.3. The water content for humid IL and salt-in-humid IL electrolytes in experiment.

		salt-water ratio	water content (ppm)
[Pyr ₁₃][TFSI]	Humid IL	0	4474
	Salt-in-humid IL	1:1	4157
		2:1	3895
		4:1	3460
[Bmim][TFSI]	Humid IL	0	5326
	Salt-in-humid IL	1:1	4901

It should be noted that, 1) the water contents of humid RTILs are 4392 ppm (MD simulation) and 4474 ppm (CV measurements) for $[\text{Pyr}_{13}][\text{TFSI}]$, and 5168 ppm (MD simulation) and 5326 ppm (CV measurements) for $[\text{Bmim}][\text{TFSI}]$, respectively; 2) the water-IL ratio in humid IL is equal to that in salt-in-humid IL, and the smaller water

concentration in salt-in-humid IL is because of the increase in total mass of electrolyte after adding salt.

As for the comment on the solubility limits of LiTFSI in [Pyr₁₃][TFSI], Henderson *et al.*¹⁶ studied the phase diagrams for the mixtures between LiTFSI and [Pyr₁₃][TFSI]. As shown in **Figure R1.7**, the melting point of LiTFSI-[Pyr₁₃][TFSI] binary mixture is lower than 21 °C when the ratio of salt to IL is lower than 3:7 (i.e., 0.3LiTFSI-0.7[Pyr₁₃][TFSI]).¹⁶ “*The binary mixtures do not crystallize for several days when stored at ambient temperature*”.¹⁶ In our work, the salt-in-humid [Pyr₁₃][TFSI] solutions with different salt-IL ratios all have the salt-IL ratio smaller than 3:7.

[REDACTED]

Figure R1.7 | (x)LiTFSI-(1-x)[Pyr₁₃][TFSI] phase diagram. This figure is copied from Ref. 16.

To experimentally check whether the solubility limit of salt in RTILs has been reached in our salt-in-humid RTIL electrolytes, we prepared the salt-in-humid RTIL electrolytes by dissolving LiTFSI into humid [Pyr₁₃][TFSI] and [Bmim][TFSI] with a molar salt-water ratio of 4:1 (the highest salt-water ratio in our work), and obtained solutions of 0.276LiTFSI-0.069H₂O-0.655[Pyr₁₃][TFSI] and 0.296LiTFSI-0.074H₂O-0.630[Bmim][TFSI]. **Figure R1.8** shows that no undissolved crystal is observed in the vessel. In other word, solubility limits of LiTFSI in such two RTILs are not

approached for the electrolytes used in this work.

Figure R1.8 | Images of pure ILs and salt-in-humid ILs. No undissolved crystal is observed.

Brief summary:

Following the reviewer’s constructive suggestion, more detailed information on the comparison of the similarities on corresponding concentrations between the experiment and the simulation, and the solubility limits of Li⁺ salt have been added into the revised manuscript (*Page 3, lines 56-57; Page 5, lines 125-126; Page 14, lines 401-403*; and the caption of **Fig. 2**). Details of related MD simulation information and preparation for humid IL and salt-in-humid IL electrolytes have been incorporated into **Supplementary Table 1** (*Page 3, lines 34-40*), and **Supplementary Table 2** (*Page 5, line 77*), respectively. The experimental examination for the salt solubility in ILs has been incorporated into **Supplementary Information**, as **Supplementary Note 2** (*Page 6, lines 78-91*) with **Supplementary Figure 3**.

Specifically, the modification in main text is listed as:

“Detailed information of simulation can be referred to **Methods** and **Supplementary Table 1**.” (*Page 3, lines 56-57*)

“As shown in **Fig. 2c**, when [Pyr₁₃][TFSI] becomes humid (4474 ppm water, close to 4392 ppm adopted in MD simulation), its electrochemical window is obviously narrowed down....” (*Page 5, lines 125-126*)

“Even when the molar salt-water ratio reaches 4:1 (the highest used in this work), the solubility limit of LiTFSI in such two RTILs still does not occur in experiment

(Supplementary Figure 3).⁶⁸” (Page 14, lines 401-403)

“Water contents for humid [Pyr₁₃][TFSI] RTILs are 4392 ppm and 4474 ppm for MD simulation and cyclic voltammograms experiments, respectively.” (caption of Fig. 2)

New reference for the solubility limit of LiTFSI in IL has been added, as:

“68 Henderson, W. A. & Passerini, S. Phase Behavior of Ionic Liquid–LiX Mixtures: Pyrrolidinium Cations and TFSI Anions. *Chemistry of Materials* 16, 2881-2885 (2004).”

Comment 5) Lines 92-93: The authors show that the affinity of Li⁺ ions for adsorption on the negatively charged electrode surface is higher than for ionic liquid cations. Please consider explaining this phenomenon in more detail. What is the proportion of Li⁺ coordinated to water molecules and Li⁺ remaining in association with the TFSI anions? Can this be better quantified? Also, how is the water arranged in this situation.

Response:

We thank the reviewer for these very insightful comments.

For the description in Lines 92-93: “As the voltage becomes more negative, owing to the strong electrostatic interaction with the charged electrode, the Li⁺ ions could overcome the energy barrier by RTIL ion layer and then be attracted to the electrode”. We would explain it this in detail: under moderate negative polarization, there is an anion twin-peak (0.4 ~ 0.7 nm) and another ion peak located at ~0.9 nm, which both attract Li⁺ ions and make them stay away from electrode surface. Under high negative polarization, the anions become obviously fewer and form one peak (either a twin peak or single peak) with much smaller peak height, which would reduce the attraction of Li⁺ ions; meanwhile, the coulombic interaction between Li⁺ ions and negative electrode is increased. Therefore, Li⁺ ions become adsorbed on the electrode under high negative polarization.

As suggested, we quantified the interfacial Li⁺ associated with surrounding ionic liquids and water, based on number density profile of Li⁺ with anion [TFSI]⁻ and

water. That is, Li^+ is considered to be associated with $[\text{TFSI}]^-$, when the $\text{Li}^+ - [\text{TFSI}]^-$ distance is smaller than the radius of the solvation shell (the first valley of the number density profile of $\text{Li}^+ - [\text{TFSI}]^-$, see **Figure R1.9a**); similarly, Li^+ is considered to be associated with water, when the $\text{Li}^+ - \text{water}$ distance is smaller than the radius of the Li^+ solvation shell. It can be seen in **Figure R1.9b** that interfacial Li^+ ions are mainly associated with $[\text{TFSI}]^-$, and some interfacial Li^+ ions could be associated with water. One snapshot of association of Li^+ with $[\text{TFSI}]^-$ and with water at the negative electrode (-2 V) is shown in **Figure R1.9c**.

Figure R1.9 | The association of Li^+ with its neighbors. **a**, The number density profile of Li^+ with anion $[\text{TFSI}]^-$ and with water from bulk MD simulation with a molar salt-water ratio of 1:1. **b**, The number density of Li^+ associated with anion $[\text{TFSI}]^-$ and with water in the interfacial region. **c**, The snapshot of association of Li^+ with anion $[\text{TFSI}]^-$ and with water at the negative electrode (-2 V).

As for the water in salt-in-humid RTIL system, as shown in **Fig. 2a** of the main text, water is intended to be pulled away from the electrode surface under moderate negative polarization. Nevertheless, under high voltage, some water molecules can be attracted on the electrode surface. The association of interfacial water with Li^+ is analyzed quantitatively, that is, the proportion of water in bound state is found to increase with the polarization (e.g., 66.38% at -2 V and 99.99% at -3 V, **Table R1.4**), indicating that interfacial water molecules are mostly bound with Li^+ .

Table R1.4 | Proportion of bound water in salt-in-humid RTIL electrolyte in interfacial region at negative electrode.

	-3 V	-2.5 V	-2 V	-1.5 V	-1.0 V	-0.5 V	0 V
Free water	99.99% ±0.01%	88.31% ±3.36%	66.38% ±10.57%	-	-	-	79.52% ±8.88%

Note: Water is depleted in the interfacial region under negative polarization of -1.5 ~ -0.5 V.

Brief summary:

With the new data and discussion, the manuscript has been modified in the main text (*Page 4, lines 95-106*). Meanwhile, the association of interfacial Li⁺ with its neighboring and the proportion water in bound state in salt-in-humid IL in interfacial region at negative electrode has been incorporated into the revised **Supplementary Note 6** with **Supplementary Figure 11** and **Supplementary Table 3**. (*Pages 13-14, lines 141-165*).

Specifically, the modification in main text is listed as:

“This elimination could be ascribed to the Li⁺-water association which could be evidenced by the similarly-located peaks in the number density distributions of Li⁺ ions and water molecules (**Supplementary Figure 7**). As the voltage becomes more negative, the anions become fewer and further away from the electrode (see the peak height and location of anion distribution in **Supplementary Figure 5**), so that Li⁺ ions get reduced attraction from anions; meanwhile, owing to the stronger electrostatic interaction with the charged electrode ... The adsorbed Li⁺ ions are found to associate with [TFSI]⁻ and/or water (**Supplementary Figure 11**); the adsorbed water molecules are almost bound with Li⁺, and the proportion of water in bound state is found to increase with the polarization (e.g., 66.38% at -2 V and 99.99% at -3 V, **Supplementary Table 3**), so that free water becomes depleted near the negative electrode (middle panel of **Fig. 2a**).” (*Page 4, lines 95-106*)

Comment 6) Given the discussion so far, water is dragged to the negative interface

with the lithium, but it does not react, because the Li^+ binds it strongly. Vice versa is true for the positive polarisation, yet the positive polarisation shows less impact. Why? As the devils advocate I will argue that the water may easily split into a proton and a hydroxide (remaining with the lithium), so in the end, if the authors would use reactive force fields, the reaction may even become more efficient at negative potentials, due to water coordination with lithium at the interface. Please comment on this, and how a reactive force field may alter the results.

Response:

We appreciate the reviewer’s thoughtful comments. First, we would like to show the enhancement of electrochemical window at negative and positive sides, based on the updated CV curves in **Figure R1.10** (Thanks to the reviewer’s **Major Comment 12** on the use of reference electrode, we obtained the OCP of Ag quasi-reference electrode vs. Ag/AgCl electrode in different solutions and then corrected corresponding CV curves, and the CV curves are updated).

Figure R1.10 | Effect of adding salt on the electrochemical performances. Cyclic voltammograms of HOPG in pure $[\text{Pyr}_{13}][\text{TFSI}]$, humid $[\text{Pyr}_{13}][\text{TFSI}]$ and humid $[\text{Pyr}_{13}][\text{TFSI}]$ with adding salt. Scan rate: 100 mV s^{-1} . Water content in humid $[\text{Pyr}_{13}][\text{TFSI}]$ for experiments: $\sim 4474 \text{ ppm}$. The electrochemical window is defined by current density of $30 \mu\text{A cm}^{-2}$.

The electrochemical window is defined by current density of $30 \mu\text{A cm}^{-2}$. It can be seen that after adding salt, the electrochemical window is clearly widened (from $-1.38 \sim 1.01 \text{ V}$ to $-2.08 \sim 1.14\text{V}$), compared with pure $[\text{Pyr}_{13}][\text{TFSI}]$ (the anodic voltage limit is 1.15 V , see **Figure R1.10**). Although the enhancement at positive polarization

is 0.13 V, which is quite smaller than that at negative side (0.7 V), it shows a nearly full recovery of positive polarization. Meanwhile, similar with [Pyr₁₃][TFSI], the electrochemical window of salt-in-humid [Bmim][TFSI] is enhanced (with increases of 0.51 V and 0.21 V, respectively, for negative and positive polarizations), compared with humid [Bmim][TFSI], also exhibiting a nearly full recovery at positive polarization compared with pure [Bmim][TFSI]. Therefore, although the enhancement of negative polarization is larger than that of positive polarization in absolute value, considering that the positive polarization has a nearly full recovery, it is hard to judge which polarization would have more impact from adding salt.

Under negative polarization, based on the response to **Major Comment 1**, DFT and MD modeling have shown that adding salt could make the water electrolysis become more difficult at negative polarization, due to the decrease of interfacial water, reinforcement of O-H bond and water re-arrangement. These three factors *also* help to make OER more difficult to occur under positive polarization. Moreover, the lowered HOMO level of bound water could make the electrons of oxygen atom in bound water shift towards Li⁺ ion so that OER occurs under a higher potential. Therefore, from the mechanism understanding by these factors revealed by DFT and MD simulations, it is hard to say which polarization would have larger impact by adding salt.

As mentioned in the response to **Major Comment 1**, although the electrochemical window increased at both negative and positive polarizations by adding salt could be explained by DFT calculations and MD simulations with standard force fields, we fully agree with the reviewer that the molecular simulation with reactive force fields would directly describe the processes of chemical bond breaking/forming, and may be a quantitative prediction of electrochemical window expansion.

Brief summary:

Following the reviewer's suggestion, the above discussions have been modified in the revised manuscript (*Page 5, lines 126-130; Page 12, lines 323-327*), as:

“shrinking both cathodic and anodic voltage limits (from -2.33 ~ 1.15 V to -1.38 ~ 1.01 V) ... the electrochemical window is clearly widened (-2.08 ~ 1.14 V), especially showing a nearly full recovery of positive polarization, compared with pure [Pyr₁₃][TFSI].” (Page 5, lines 126-130)

“Although the above understanding is revealed by classical MD simulation joint with DFT calculations, and could qualitatively explain the expansion of electrochemical window, the molecular simulation with reactive force fields, which could directly mimic the processes of chemical bond breaking/forming, may give a quantitative prediction of electrochemical window expansion.” (*Conclusion & Discussion Section* of main text, Page 12, lines 323-327)

Comment 7) Line 126: The correlation of measured and simulated IR data is completely unclear. Figure 10 in the SI shows very little ratio change, and it is unclear what the black solid line refers to. A ratio plot for all peaks will be much better to understand the data and how it compares. Please provide such a plot.

Response:

We appreciate the reviewer for the constructive comment on the IR data. We're sorry that we did not give a clear description previously, including that in caption. Actually, the back line in previous **Supplementary Figure S10** represents one type of water in electrolyte. To make it clear and easier comparison with modeling results in **Fig. 3a**, the measured IR data is re-drawn as **Figure R1.11**, exactly following the format of **Fig. 3a** in the main text.

Generally, the spectrum of O-H stretching vibration can be decomposed into some individual peaks. Typically, each spectrum could be separated into three peaks, indicating that three different kinds of water are present.^{17,18} With the same approach, in our work, three Lorentz peak functions are taken to decompose the spectrum of O-H stretching vibration in **Figure R1.11**. The red, pink and blue regions represent three different kinds of water in electrolyte. It is common and expected that the predicted vibration position has some deviation from the experimental result.¹⁹

Therefore, in order to better understand the IR spectra, we, herein, concentrate on the shift of the stretching vibration peak rather than the absolute value. Taking the peak position of stretching vibration of water in humid IL as reference, the comparison of peak shifts of IR spectra from MD simulation and experiment has been plotted in **Figure R1.12**, which reveals that, despite the numerical differences, the blue-shifts due to adding salt are observed both in MD simulation and experiment.

Figure R1.11 | Experimental IR spectra in humid [Pyr₁₃][TFSI], as well as humid [Pyr₁₃][TFSI] with different molar salt-water ratios (from 0.5 to 4). The red, pink, and blue regions are the first, second, and third fitting spectrum, respectively, each representing one type of water. The cyan dash lines and gray solid lines represent the summation of the fitting spectra and the experimental spectra.

Figure R1.12 | Comparison of MD calculated and experimental peak position shifts of the O-H stretching vibration relative to humid RTIL system. The spectrum of O-H stretching vibration is assigned into three different models. Taking the peak position of O-H stretching vibration of water in humid IL as reference, $\Delta\nu$ is the peak position shift of O-H stretching vibration of water in salt-in-humid RTIL.

Brief summary:

The discussions on the IR data have been incorporated into the revised manuscript (*Page 6, lines 146-148; Page 6, lines 157-161*). The comparison of calculated and experimental frequency shifts can be found in the updated **Supplementary Note 9** with **Supplementary Figures 15-16** (*Pages 18-19, lines 223-247*).

Specifically, the modification in main text is listed as:

“To better delineate the change of the stretching vibration of water, three Lorentz peak functions, with each peak representing one type of water, are taken to decompose the IR spectra.” (*Page 6, lines 146-148*)

“The peak location shifts of stretching vibrations were shown in **Supplementary Figure 16**, compared with those obtained from MD simulation in **Fig. 3a**. It can be seen that in spite of the numerical difference, the blue-shifts (i.e., the peak location of stretching vibration shifts towards higher wavenumber) occur after adding salt in both simulation and experiment.” (*Page 6, lines 157-161*)

Comment 8) The discussion of the IR data is also quite confusing at not in line with the plots. Please consider rewriting. E.g. please describe qualitatively, why the coordination of Li⁺ to water is preferred over the coordination of Li⁺ to IL anions? Also include a list of peak shifts and relative comparison of peak shifts of experiment and theory.

Response:

We appreciate the reviewer for the suggestion.

The discussion of the IR data is rephrased in the manuscript. Briefly, to better delineate the change of the stretching vibration of water, three Lorentz peak functions, with each peak representing one type of water, are taken to decompose the IR spectra. The peak location shifts of stretching vibrations were shown in **Figure R1.12**, compared with those obtained from MD simulation in **Fig. 3a**. It can be seen that in spite of the numerical difference, the blue-shifts occur after adding salt both in simulation and experiment. The blue-shift is attributed to the destroyed hydrogen

bond (H-bond) network.^{18,20} Therefore, H-bonds between water molecules and between water molecules and RTIL ions were computed as a function of the salt-water ratio (**Fig. 3b**), showing that H-bonds are reduced by adding salt.

Based on our analysis of interaction energy between water and surroundings (including Li^+ , water, cation $[\text{Pyr}_{13}]^+$ and anion $[\text{TFSI}]^-$), we found that the interaction between water and Li^+ is the strongest (**Fig. 3c-d**). This indicates that the coordination of Li^+ to water is preferred over that of water to ILs (both cations and anions), leading to the destruction of H-bond network.

We agree with the reviewer that plot for peak shifts of O-H stretching vibrations should be added for a relative comparison of experiment and theory. The relative comparison of peak shifts obtained from experiment and theory has been shown in **Figure R1.12**. In spite of the numerical differences, the blue-shifts occur with adding salt both in MD simulations and our experiments.

Brief summary:

The discussions on the IR data have been incorporated into the revised manuscript (*Page 6, lines 151-163; Pages 6-7, lines 168-175*). The comparison of calculated and experimental frequency shifts can be found in the updated **Supplementary Note 9** with **Supplementary Figures 15-16** (*Pages 18-19, lines 223-247*).

Specifically, the modification in main text is listed as:

“Compared with pure water, IR spectra of humid $[\text{Pyr}_{13}][\text{TFSI}]$ show occurrence of new peaks at higher wavenumber (see the bottom two panels of **Fig. 3a**), which could be ascribed to the reduction of the water clusters by the existed RTIL, in accord with previous work.^{39,40} With adding salt, the peak positions of stretching vibrations shift towards even higher wavenumber when the salt-water ratio varies from 0 to 0.5 then to 4. The IR spectra of O-H stretching vibrations were measured by Fourier transform infrared spectroscopy (FTIR) for humid $[\text{Pyr}_{13}][\text{TFSI}]$ with adding salt (**Supplementary Figure 15**). The peak location shifts of stretching vibrations were shown in **Supplementary Figure 16**, compared with those obtained from MD

simulation in **Fig. 3a**. It can be seen that in spite of the numerical difference, the blue-shifts (i.e., the peak location of stretching vibration shifts towards higher wavenumber) occur after adding salt in both simulation and experiment. Moreover, IR spectra of O-H stretching vibrations were also obtained from DFT calculations for humid [Pyr₁₃][TFSI] (**Supplementary Figure 17**), which confirms the occurrence of the blue-shift with adding Li⁺ ion.” (Page 6, lines 151-163)

“The blue-shift is attributed to the destroyed hydrogen bond (H-bond) network.^{37,39} Therefore, H-bonds between water molecules and between water molecules and RTIL ions, defined by the geometrical criterion in **Supplementary Figure 18**, were computed as a function of the molar salt-water ratio (**Fig. 3b**), showing that H-bonds are reduced by adding salt. Especially, those between water molecules nearly disappear with the molar salt-water ratio over 1.5:1, which can be also evidenced by the ... (**Supplementary Figure 19**).” (Pages 6-7, lines 168-175)

Comment 9) The authors show that Li⁺ ions preferentially stay away from the electrode surfaces but in lines 92-93 the authors say that the Li⁺ ions are attracted to the electrode surface. Please clarify. This also applies to discussion in lines 247-249. The location of Li⁺ ions as well as their coordination to water molecules should be discussed with a proper scrutiny and orderly throughout the paper.

Response:

We appreciate the reviewer for these comments. The discussion on the preferential location of Li⁺ in the previous manuscript may be misleading, since we did not clarify the corresponding voltage range. We modified the discussions with the specific voltage range, using terminology of moderate and high negative polarizations.

Discussed in the above response to **Major Comment 5**, under moderate negative polarization (-1.5 ~ 0 V), Li⁺ ions preferentially stay away from the electrode surfaces, locating at around 0.4 to 0.7 nm. However, under high negative polarization (more negative than -1.5 V), the anions become obviously fewer and form one peak (either a twin peak or single peak) with much smaller peak height, which would reduce the

attraction with Li^+ ions; meanwhile, the coulombic interaction between Li^+ ions and negative electrode is increased. Therefore, Li^+ ions become adsorbed on the electrode under high negative polarization.

The association of Li^+ with water and $[\text{TFSI}]^-$ can be seen in **Figure R1.9** that interfacial Li^+ ions are mainly associated with $[\text{TFSI}]^-$, and some interfacial Li^+ ions could be associated with water.

Brief summary:

The descriptions are rephrased with a proper way, and modified in the main text (*Page 4, lines 95-106; Page 11, lines 310-311*). Meanwhile, the association of interfacial Li^+ with its neighboring and the proportion water in bound state in salt-in-humid RTIL electrolyte in interfacial region at negative electrode has been incorporated into the revised **Supplementary Note 6** with **Supplementary Figure 11** and **Supplementary Table 3**. (*Pages 13-14, lines 141-165*)

Specifically, the modification in main text is listed as:

“This elimination could be ascribed to the Li^+ -water association which could be evidenced by the similarly-located peaks in the number density distributions of Li^+ ions and water molecules (**Supplementary Figure 7**). As the voltage becomes more negative, the anions become fewer and further away from the electrode (see the peak height and location of anion distribution in **Supplementary Figure 5**), so that Li^+ ions get reduced attraction from anions; meanwhile, owing to the stronger electrostatic interaction with the charged electrode ... The adsorbed Li^+ ions are found to associate with $[\text{TFSI}]^-$ and/or water (**Supplementary Figure 11**); the adsorbed water molecules are almost bound with Li^+ , and the proportion of water in bound state is found to increase with the polarization (e.g., 66.38% at -2 V and 99.99% at -3 V, **Supplementary Table 3**), so that free water becomes depleted near the negative electrode (middle panel of **Fig. 2a**).” (*Page 4, lines 95-106*)

“water is intended to be pulled away from the electrode under positive and moderately negative polarizations....” (*Page 11, lines 310-311*)

Comment 10) Line 239 Discussion: Have similar effects of salt addition on the expansion of electrochemical windows in IL systems been discussed in previous works? The discussion does not currently include a proper 'state-of-the-art' related to this topic. Also the work may benefit from expanding the discussion section by including some more detailed insight from literature. Please consider also enriching the discussion by referring to the experimental results from similar systems.

Response:

We appreciate the reviewer's suggestion.

Indeed, the behavior of water in humid IL electrolytes at electrified interfaces has been studied with molecular simulations and experiments.²¹⁻²³ Ionic liquids, when mixed with water, were found to have reduced electrochemical window, which would be attributed to the water adsorbed on electrodes in humid hydrophobic RTIL electrolytes. Therefore, a key task is how to enhance the narrowed electrochemical window of humid RTIL electrolytes, by minimizing the electrosorption of water on the electrode surfaces or lowering the activity of adsorbed water. We shall admit that before this work, there is no study showing that adding salt could expand the electrochemical windows of humid RTILs.

In this work, using both simulation and experiment, we find this convenient approach of adding salt to expand the electrochemical window of humid hydrophobic RTILs, and understand such expansion mechanism by four factors: (1) *decrease of interfacial water*, (2) *arrangement change of interfacial water*, (3) *reinforcement of O-H bond in bound water*,^{3,24,25} and (4) *lowered HOMO level of bound water*,¹ by the combination of DFT calculations and MD simulations. The mitigation of the adsorbed water and its impact could improve the practical performance of electrochemical energy storage systems, as the electrochemical window of humid RTILs can be fully exploited, and may also benefit other applications such as RTIL gating²⁶, lubrication²⁷, and electrowetting²⁸. The understanding of adding salt to expand the electrochemical window, including the decrease of interfacial water, reinforcement of O-H bond and water re-arrangement as well as the lowered HOMO level in bound water, may be responsible for water-in-salt electrolytes^{1,2,5,29-31} in battery and supercapacitor.

It should be noted that the approach of adding salt is only tested in two hydrophobic RTILs herein, and taking into account the cost of lithium salt, more RTILs and cheap salts^{32,33} preferred by further investigation.

Brief summary:

More discussions have been integrated in the revised manuscript (*Page 12, lines 327-329; Page 12, lines 334-339*) as:

“Moreover, it should be noted that the approach of adding salt is tested in two hydrophobic RTILs herein, and taking into account the cost of lithium salt, more RTILs and cheap salts^{46,47} are preferred by further investigation.” (*Page 12, lines 327-329*)

“... and may also benefit other applications such as RTIL gating⁴⁸, lubrication⁴⁹, and electrowetting⁵⁰. The understanding of adding salt to expand the electrochemical window, including the decrease of interfacial water, reinforcement of O-H bond and water re-arrangement as well as the lowered HOMO level in bound water, may be responsible for the working mechanism of water-in-salt electrolytes²¹⁻²⁶ in battery and supercapacitor.” (*Page 12, lines 334-339*)

New references on the discussion are added, as:

- 46 Lukatskaya, M. R. et al. Concentrated Mixed Cation Acetate “Water-in-Salt” Solutions as Green and Low-Cost High Voltage Electrolytes for Aqueous Batteries. *Energy & Environmental Science* 11, 2876-2883 (2018).
- 47 Dou, Q. et al. A Sodium Perchlorate-Based Hybrid Electrolyte with High Salt-to-Water Molar Ratio for Safe 2.5 V Carbon-Based Supercapacitor. *Energy Storage Materials* 23, 603-609 (2019).
- 48 Zhao, S. et al. Quantitative Determination on Ionic-Liquid-Gating Control of Interfacial Magnetism. *Advanced Materials* 29, 1606478 (2017).
- 49 Fajardo, O. Y., Bresme, F., Kornyshev, A. A. & Urbakh, M. Water in Ionic Liquid Lubricants: Friend and Foe. *ACS Nano* 11, 6825-6831 (2017).
- 50 Millefiorini, S., Tkaczyk, A. H., Sedev, R., Efthimiadis, J. & Ralston, J. Electrowetting of Ionic Liquids. *Journal of the American Chemical Society* 128, 3098-3101 (2006).”

Comment 11) In many places, the discussion of the results is rather qualitative than quantitative. I mean using expressions such as: more interaction, more or less bound, etc. Can the authors review their discussion of the results in more quantitative manner using coordination numbers, bond energies? Can the authors also quantitatively discuss the occupancy of species near the charged surfaces in terms of ratios of different species Li:water:IL?

Response:

We thank the reviewer for the useful advice. The quantitative discussions have been incorporated into the revised manuscript (*Page 7, lines 178-181; Page 7, lines 185-187; Page 6, lines 166-168; Page 4, lines 102-106*).

Regards to **interaction, and more or less bound**, we revise as:

“As presented in **Fig. 3c**, for humid RTIL, the interaction between water and anions is pronounced (about -34 kJ mol^{-1}), followed by that between water and cation (*ca.* -20 kJ mol^{-1}) and between water and water (around -10 kJ mol^{-1}).” (*Page 7, lines 178-181*)

Regards to **coordination numbers**, we revise as:

“Meanwhile, the coordination number of Li^+ ions around water increases from about 0.61 to 0.95 as the molar salt-water ratio varies from 0.5:1 to 4:1 (**Supplementary Figure 20**).” (*Page 7, lines 185-187*)

The figure of coordination number of Li^+ associated with water within the hydration shell as a function of the molar salt-water ratio is incorporated into the revised **Supplementary Note 9**, with **Supplementary Figure 20** (*Page 22, lines 294-305*).

Regards to **bond energies** of O-H, we describe them with a qualitative way based on previous work,^{3,24,25} as:

“Essentially, the blue-shift indicates that more energy is required to cleave the O-H bond of water into radicals, and the O-H bond is reinforced, suggesting that with adding salt, water has a lower activity and thus its electrochemical decomposition needs a higher potential⁴¹⁻⁴³.” (*Page 6, lines 166-168*)

The related references have newly been added to support this conclusion, as:

- “41 Badger, R. M. The Relation Between the Energy of a Hydrogen Bond and the Frequencies of the O–H Bands. *The Journal of Chemical Physics* 8, 288-289, (1940).
- 42 Zavitsas, A. A. Quantitative relationship between bond dissociation energies, infrared stretching frequencies, and force constants in polyatomic molecules. *The Journal of Physical Chemistry* 91, 5573-5577, (1987).
- 43 Xie, J., Liang, Z. & Lu, Y.-C. Molecular crowding electrolytes for high-voltage aqueous batteries. *Nature Materials*, (2020), doi.org/10.1038/s41563-020-0667-y.”

Regards to occupancy of species near the charged surfaces, we change as:

“The adsorbed Li^+ ions are found to associate with $[\text{TFSI}]^-$ and/or water (**Supplementary Figure 11**); the adsorbed water molecules are almost bound with Li^+ , and the proportion of water in bound state is found to increase with the polarization (e.g., 66.38% at -2 V and 99.99% at -3 V, **Supplementary Table 3**), so that free water becomes depleted near the negative electrode (middle panel of **Fig. 2a**).” (Page 4, lines 102-106)

Additionally, the quantitative association of interfacial Li^+ with surrounding ionic liquids and water has been incorporated into **Supplementary Note 6**, with **Supplementary Figure 11** (Page 13, lines 141-154).

Comment 12) In Figure 2c and where-else appropriate, please correlate the silver reference potential to standard Ag/AgCl or any other standard reference potential (also please mention these parameters in experimental section). I also doubt the choice of this electrode. It is not very stable in the solutions used in this work. Please provide an OCP of this electrode in the different solutions, and also correct the CVs accordingly. What is the error of this electrode in terms of the reference potential error? This is important.

Response:

We appreciate the reviewer’s constructive comment. We agree with the reviewer that the potential of Ag wire may not be as stable as that of Ag/AgCl reference electrode. The reason why we selected Ag wire is to avoid possible contamination due to the

leakage of Ag/AgCl reference electrode. With the same reason, for investigating the interface between electrode and ionic liquids, metal wire was frequently used as reference electrode.³⁴⁻³⁷ For example, Prof. Dieter M. Kolb group conducted their experiments using metal wire reference electrode.³⁷

Furthermore, following the reviewer's suggestion, the time dependence of the OCP of Ag quasi-reference electrode vs. Ag/AgCl electrode in different solutions is measured. As shown in **Figure R1.13**, the time-evolved OCP of Ag quasi-reference electrode fluctuated a little bit. The OCP and error bar were listed in **Table R1.5**. Meanwhile, the CV curves are updated in **Figure R1.10**, accordingly based on OCPs of Ag wire versus the Ag/AgCl reference electrode in different electrolytes.

Figure R1.13 | Time-dependence of open-circuit potential (OCP). a-b, Time-dependence of OCP of Ag quasi-reference electrode vs. Ag/AgCl electrode in [Pyr₁₃][TFSI], humid [Pyr₁₃][TFSI] and salt-in-humid [Pyr₁₃][TFSI] (a) and in [Bmim][TFSI], humid [Bmim][TFSI] and salt-in-humid [Bmim][TFSI] (b).

Table R1.5. The open-circuit potential (OCP) and error of Ag wire versus the Ag/AgCl reference electrode in different electrolytes. Unit: V

	pure IL	humid IL	with salt
[Pyr ₁₃][TFSI]	-0.24±0.03	-0.19±0.05	-0.24±0.04
[Bmim][TFSI]	0.00±0.04	0.08±0.05	0.03±0.04

Brief Summary:

The above discussion has been incorporated into the revised manuscript in *Experimental materials and measurements Section* (Page 15, lines 413-417). Furthermore, detailed information about the time-dependence of the open-circuit potential of Ag quasi-reference electrode vs. Ag/AgCl electrode in different electrolytes has been updated in the **Supplementary Note 7**, with **Supplementary Figure 12** and **Supplementary Table 4** (Page 15, lines 166-180).

Specifically, the modification in main text is listed as:

“The potentials of Ag wire in all electrolytes were calibrated with respect to an Ag/AgCl reference electrode. The time-dependence of the open-circuit potential (OCP) of Ag quasi-reference electrode vs. Ag/AgCl electrode in different solutions is measured (**Supplementary Figure 12**), and such OCPs for different electrolytes were listed in **Supplementary Table 4.**” (Page 15, lines 413-417)

Minor concerns and typos

Comment 1) Please consider to add a 'Conclusions' section

Response:

We thank the reviewer for this helpful advice. Following this and considering we added more discussions, such section is renamed as “**Conclusion & Discussion**”.

Comment 2) A general concern: Please clarify what "bound water" is by definition in the different situations described in the manuscript.

Response:

We thank the reviewer for pointing out the definition of “bound water”.

Previous studies have tried to describe the influence of Li⁺-salt on the electrolysis of water in battery community, where water is divided into “free” and “bound”.^{1,29} For instance, Suo *et al.*²⁹ quantified the fraction of “free” water as “*which is not bound to any Li⁺ within the primary solvation sheath of 0.27 nm.*” McEldrew *et al.*³⁸ divided

water into free water and bound water based on the correlations between water and Li^+ , as “*The strong correlations between cations and water permits the partitioning of water into **bound** and **free** states*”, and “*we assume all water molecules with the first shell of Li^+ to be **bound** to Li^+* ”. With this definition, they quantified the distribution of free water and bound water along with EDL as a function of the voltages.³⁸ In our work, we adopt the same definition of free and bound water.

Brief Summary:

Based on the above discussions, our work adopts the same definition throughout the manuscript (*Page 4, lines 89-92*), shown as,

“We divide the **remaining** adsorbed water into free and bound states, based on the number density profile of Li^+ ions around water molecule (i.e., water is considered to be bound to Li^+ within a distance of around 0.25 nm, otherwise it is labeled as free water, see **Supplementary Figure 10**).^{21,24,27}” (*Page 4, lines 89-92*)

Comment 3) Figure 1: It will be nice to flip x/y axis for all panels, to enhance comparison with CV data (see also Figure 2 where this is the case).

Response:

We thank the reviewer for this constructive suggestion, and have modified it as,

Fig. 1 | Ion and water distributions under various voltages. a-c, Number densities of cation (**a**), anion (**b**), and water (**c**) in humid [Pyr₁₃][TFSI] as a function of distance from the electrode. **d-f,** The number densities of cation (**d**), anion (**e**), and water as well as Li⁺ (**f**), in salt-in-humid [Pyr₁₃][TFSI]. The **horizontal** dash lines ($z = 0.35$ nm) in (**c**) and (**f**) represents the **upper** boundary of the interfacial region. The contour in (**f**) indicates number densities of Li⁺ ions. Unit: # nm⁻³. The molar **salt-water** ratio is 1:1.

Comment 4) Figure 1: Please discuss why ions are generally further away around the potential of zero charge.

Response:

We thank the reviewer for this comment. Our MD simulations revealed that the EDL consists of cation and anion layers oscillating up to few nanometers from the electrode surfaces, with cations being closer to the charged surface for negative potentials and anions for positive potentials, compared with the ion distribution at PZC. Specifically, the first ion layer for [Pyr₁₃]⁺ and [TFSI]⁻ is located at ~0.42 nm and ~0.43 nm, respectively, at PZC. As the electrode becomes positively charged, [Pyr₁₃]⁺ ions are getting away from the electrode, leaving more space for anions to be fit in; meanwhile, due to the increased anion-electrode coulombic interactions, anions would move closer to the electrode surface (from ~0.43 nm at PZC to ~0.40 nm at 3 V). This phenomenon has also been demonstrated in other work by MD simulation and atomic force microscopy (AFM) experiments.³⁹⁻⁴¹ As shown in **Figure R1.14** coulombic interactions dominate the interaction between ions and the electrode under the voltage of 1 V.³⁹ The similar explanation holds for the cations getting closer to the negatively charged electrode, compared with PZC.

[REDACTED]

Figure R1.14 | MD calculation of the ion-electrode interaction at PZC and 1 V.³⁹ IL is [Emim][TFSI]. This figure is copied from Ref. 39.

Brief Summary:

The above discussion has been incorporated into the revised manuscript (*Page 3, lines 69-72*), as:

“One can find that as the electrode is polarized, the counterion, $[\text{Pyr}_{13}]^+$ for the negative electrode and $[\text{TFSI}]^-$ for the positive electrode, moves closer to the charged surface, in consistence with previous studies,^{29,30} which could be understood by the increased counterion-electrode coulombic interactions.^{29,30}”

New references are added, as:

“29 Black, J. M. et al. Bias-Dependent Molecular-Level Structure of Electrical Double Layer in Ionic Liquid on Graphite. *Nano Letters* 13, 5954-5960 (2013).

30 Vatamanu, J., Borodin, O. & Smith, G. D. Molecular Insights into the Potential and Temperature Dependences of the Differential Capacitance of a Room-Temperature Ionic Liquid at Graphite Electrodes. *Journal of the American Chemical Society* 132, 14825-14833 (2010).”

Comment 5) Figure 2b is very misleading, where firstly F is not defined in the figure's caption. I assume it is a force. Presumably, the authors were trying to mark F as the repulsion force to indicate that the Li^+ cations drive the water molecules away from the surface during anodic polarization shown on the right hand side (red). However, I don't understand why the Li^+ -associated water molecule is still being driven away from the surface during the cathodic polarization in the exactly same way shown on left hand side (blue). The direction of the repulsive force and the schematic orientation of the molecule does not make sense in the second case, please clarify this.

Response:

We appreciate the reviewer for these valuable comments.

As the reviewer pointed out, we wish to express that the water molecules stay away from the electrode surface by adding Li^+ ions. The marker F is the repulsion force, which is right at positive side, but may give misleading at negative side. To avoid the misleading identification of marker F, we deleted it in the revised figure.

As for the orientation of water molecules, we have checked the orientation distribution of water based on MD simulation (see **Figure R 1.5**), and re-plotted the

Schematic in **Fig. 2b**.

Brief summary:

Following the reviewer's advice, we have re-plotted the Schematic in **Fig. 2b**, as

Fig. 2 | Effect of adding salt on the interfacial water and voltage window. a, Electro-sorption of water from humid [Pyr₁₃][TFSI] with/without adding salt. The top, middle and bottom panels are, respectively, the **number densities of total (ρ_n^{tot})**, **free (ρ_n^{free})** and **Li⁺-bound water (ρ_n^{bound})** in the interfacial region. **b**, Schematic of effect of adding salt on water electro-sorption. **c**, Cyclic voltammograms of HOPG in pure [Pyr₁₃][TFSI] (black line), humid [Pyr₁₃][TFSI] red line) and salt-in-humid [Pyr₁₃][TFSI] electrolyte (the **molar salt-water ratio = 1:1**, blue line). Scan rate: 100 mV s⁻¹. **The electrochemical window is defined by current density of 30 $\mu\text{A cm}^{-2}$.** Water contents for humid [Pyr₁₃][TFSI] RTILs are 4392 and 4474 ppm for MD simulation and cyclic voltammetry experiments, respectively.

Comment 6) throughout the paper: Naming 'cation' is misleading in the text and in all the figures because it can refer either to Li+ or to ionic liquid cation. Please specify this everywhere precisely.

Response:

We thank the reviewer for pointing out such a misleading description. It is true that

‘cation’ in all figures is ionic liquid cation $[\text{Pyr}_{13}]^+$ rather than Li^+ . Following reviewer’s suggestion, we have re-edited the expression of cation to be more specific terms as $[\text{Pyr}_{13}]^+$.

Comment 7) Methods section: Experimental method sections have not enough details about preparation of solutions for experimental IR measurement.

Response:

We thank the reviewer for this useful advice.

To prepare the humid IL, 6.4 μL (0.36 mmol) ultrapure water was added into dry $[\text{Pyr}_{13}][\text{TFSI}]$ (1.00 mL, $\rho=1.43 \text{ g cm}^{-3}$, 3.50 mmol), and then the mixture was stirred till homogeneous solution was formed. The molar ratio of water to IL is around 1:9.7. Then the water content was 4474 ppm, close to 4392 ppm adopted in MD simulation. As for salt-in-humid IL, 6.4 μL (0.36 mmol) ultrapure water, and 0.1033 g (0.36 mmol) vacuum dried LiTFSI, i.e., the molar salt-water ratio is 1:1, were added into dry $[\text{Pyr}_{13}][\text{TFSI}]$ (1.00 mL, $\rho=1.43 \text{ g cm}^{-3}$, 3.50 mmol), and then the mixture was stirred till homogeneous solution was formed. Correspondingly, the molar ratio for salt: water: $[\text{Pyr}_{13}][\text{TFSI}]$ is 1:1:9.7. Similarly, with the same contents of water and $[\text{Pyr}_{13}][\text{TFSI}]$, the mass of LiTFSI was 0.2066 g and 0.4132 g, respectively, to prepare the salt-in-humid IL with the molar salt-water ratio of 2:1 and 4:1. With the same approach, to prepare humid $[\text{Bmim}][\text{TFSI}]$, 7.7 μL (0.43 mmol) ultrapure water was added into dry $[\text{Bmim}][\text{TFSI}]$ (1.00 mL, $\rho=1.44 \text{ g cm}^{-3}$, 3.43 mmol), and then the mixture was stirred till homogeneous solution was formed. Thus, the molar ratio of water to IL is around 1:8 and the water content was as 5326 ppm, close to 5168 ppm adopted in MD simulation. Meanwhile, 7.7 μL (0.43 mmol) ultrapure water and 0.1234 g (0.43 mmol) LiTFSI were added into 1.00 mL $[\text{Bmim}][\text{TFSI}]$, and the mixture was stirred to form salt-in-humid IL with the molar salt-water ratio is 1:1. It should be noted that the molar water-IL ratio in humid IL is equal to that in salt-in-humid IL, and the smaller water concentration in salt-in-humid IL is because of the increase in total mass of electrolyte with adding salt. The water content for all

electrolytes used is listed in **Table R1.3**.

As for the IR measurement, Fourier-transform infrared spectroscopy (FTIR) measurements were conducted on a Nexus 8700 spectrometer (Nicolet) equipped with a liquid-nitrogen-cooled MCT-A detector. A commercial demountable liquid cell (Harrick Scientific Corporation, **Figure R1.15**) was used. The sample (RTIL-LiTFSI-water mixture) was prepared in glove-box and transferred to spectrometer with a sealed vessel. The sample preparation for IR measurements was completed within two minutes. Then, an unpolarized IR radiation sequentially passed through two CaF₂ windows with a thin-layer solution (25 μm). The IR transmission spectrum of the sample was taken from 1111 to 4000 cm⁻¹ with a resolution of 4 cm⁻¹ and averaged 100 times.

Figure R1.15 | Schematic diagram of demountable liquid cell for FTIR measurements.

Brief summary:

The preparation of solutions for experimental IR measurements has been incorporated into the revised manuscript (*Experimental materials and measurements Section, Page 14, lines 400-401; Page 15, lines 421-425*). Details of the preparation for humid IL and salt-in-humid IL electrolytes have been incorporated into the revised **Supplementary Note 2**, with **Supplementary Table 2** (*Page 5, lines 57-77*).

Specifically, the modification in main text is listed as:

“Detailed information of electrolyte preparation can be seen in **Supplementary Note 2**.” (*Page 14, lines 400-401*)

“A commercial demountable liquid cell (Harrick Scientific Corporation) was used. The sample (RTIL-LiTFSI-water mixture) was prepared in glove-box and transferred to spectrometer with a sealed vessel. The sample preparation for IR measurements was completed within two minutes. Then, an unpolarized IR radiation sequentially passed through two CaF₂ windows with a thin-layer solution (25 μm).” (Page 15, lines 421-425)

Comment 8) Line 11: lowered the activity of what?

Response:

We thank the reviewer for pointing out this confusing expression. Actually, it is the activity of “water remaining on the electrode”, and it would be corrected as “lowered the activity of the interfacial water”. However, due to the word limitation of Abstract, we have now rephrased this sentence (Page 1, line 11) as:

“lowering its activity” (Page 1, line 11)

Comment 9) Line 74: distributions – typo

Response:

We thank the reviewer for pointing out this typo, and have corrected this.

Comment 10) Line 465 (Figure 1): At a first glance, the figure labelling 'cation' may be misleading as both Li⁺ and [Pyr13]⁺ are cations; Please specify that the label 'cation' in subplots a) and d) refers only to [Pyr13]⁺. It could be also beneficial for the figure readability to highlight better that the distribution of Li⁺ is only shown in the subplot f).

Response:

We appreciate the reviewer for these valuable comments. Following the reviewer’s suggestions, we re-edit the expression of cation to more specific terms as [Pyr₁₃]⁺.

We agree with the reviewer that it would be readable to solely show the distribution of

Li⁺ in **Fig. 1f**. Since water adsorption is the key issue to determine the electrochemical window of electrolyte, and to better compare the change of the distribution of water before (panel c of **Fig. 1**) and after adding salt (panel f of **Fig. 1**), it may be better to show distributions of water and Li⁺ together (e.g., showing the how water would be close to Li⁺). However, we agree that it would be good idea to plot distributions of water and Li⁺ separately, which has been added as **Supplementary Figure 9** in the revised **Supplementary Information**.

Brief Summary:

Following the reviewer's advice, we plot the number density distribution of water and Li⁺ separately. See **Supplementary Figure 9** (Page 11, lines 125-130).

Supplementary Figure 9 | Water and Li⁺ distributions under various voltages in salt-in-humid [Pyr₁₃][TFSI]. a-b, Number densities of water (a) and Li⁺ (b) in salt-in-humid [Pyr₁₃][TFSI] as a function of distance from the electrode. The horizontal dash lines ($z = 0.35$ nm) in (a) and (b) represent the upper right boundary of the interfacial region. Unit: # nm⁻³. The molar salt-water ratio is 1:1.

Comment 11) Line 472 (Figure 2): Please explain why in humid IL (with no Li⁺ added) there is more water electrosorbed in positive polarization than in a negative polarization.

Response:

We thank the reviewer for this comment.

The above observations that “there is more water electrosorbed in positive polarization than in a negative polarization” could be understood by PMF profiles of

water molecule in [Pyr₁₃][TFSI] in **Fig. 4a** of the main text. Specifically, the free energy relative to the bulk state of water near surface is around -10 to -20 kJ mol⁻¹ at +2 V and +3 V, deeper than that at -2 V (*ca.* -5 kJ mol⁻¹) and -3 V (*ca.* -9 kJ mol⁻¹), indicating that water prefers to accumulate at positive electrodes than at negative electrodes.

Brief summary:

The above discussion has been incorporated into the revised manuscript (*Page 8, lines 210-213*), as:

“Furthermore, the free energy of interfacial water is around -10 and -20 kJ mol⁻¹, respectively, at 2 V and 3 V, deeper than that at -2 V (about -5 kJ mol⁻¹) and -3 V (around -9 kJ mol⁻¹), indicating that water would accumulate more at positive electrodes than at negative electrodes.” (*Page 8, lines 210-213*)

Comment 12) Line 472 (Figure 2): subplot a) symbol of number density ρ on y axis is not explained in the figure caption

Response:

We thank the reviewer for this advice. We have added the expression of the symbol ρ on y axis in the figure caption (caption of **Fig.2, Page 22, lines 618-619**), as:

“The top, middle and bottom panels are, respectively, the number densities of total (ρ_n^{tot}), free (ρ_n^{free}) and Li⁺-bound water (ρ_n^{bound}) in the interfacial region.”

(caption of **Fig. 2, Page 22, lines 618-619**)

Comment 13) Line 92: Typo: more 'negative' not 'negatively'

Response:

We thank the reviewer for pointing this out, and have corrected it as “**negative**”.

Comment 14) Line 95: 'more water molecules are driven into the interfacial region, but they are almost bound with Li+...' What do the authors mean by saying that the water molecules are 'almost' bound with Li+? Please clarify by using a more specific expression to describe the nature of bonding between Li+ and water molecules

Response:

We appreciate the reviewer for these comments. Following the comment, the association of interfacial water with Li⁺ is analyzed quantitatively, that is, the proportion of water in bound state is found to increase with the polarization (e.g., 66.38% at -2 V and 99.99% at -3 V, **Table R1.4**), indicating that interfacial water molecules are mostly bound with Li⁺.

As for the second comment, the description of nature of bonding between Li⁺ and water molecules has been widely used in “water-in-salt” electrolytes, where water is divided into “free” and “bound”.^{1,29,38} Details can be seen in response to **Minor Comment 2**. Therefore, we adopted the same expression as free/bound water with Li⁺ to describe the nature of bonding between Li⁺ and water molecules.

Brief summary:

The above discussion has been incorporated into the revised manuscript (*Page 4, lines 103-106*). The proportion of water in bound state in salt-in-humid IL electrolyte in interfacial region at negative electrode has been incorporated into the revised **Supplementary Note 6**, with **Supplementary Table 3** (*Page 14, lines 155-165*).

Specifically, the modification in main text is listed as:

“the adsorbed water molecules are almost bound with Li⁺, and the proportion of water in bound state is found to increase with the polarization (e.g., 66.38% at -2 V and 99.99% at -3 V, **Supplementary Table 3**), so that free water becomes depleted near the negative electrode (middle panel of **Fig. 2a**).” (*Page 4, lines 103-106*)

Additionally, based on the above discussions, our work adopts the same definition throughout the manuscript (*Page 4, lines 89-92*), shown as,

“We divide the **remaining** adsorbed water into free and bound states, based on the

number density profile of Li^+ ions around water molecule (i.e., water is considered to be bound to Li^+ within a distance of around 0.25 nm, otherwise it is labeled as free water, see **Supplementary Figure 10**).^{21,24,27,}

Comment 15) Line 472 (Figure 2): Subplot a) bottom panel. Please clarify in the figure label that the bound water refers to water bound to Li^+ cations.

Response:

We thank the reviewer for this constructive advice. Following the reviewer's suggestion, we have clarified the definition of bound water (caption of **Fig.2, Page 22, lines 618-619**) as:

“The top, middle and bottom panels are, respectively, the **number densities of total (ρ_n^{tot}), free (ρ_n^{free}) and Li^+ -bound water (ρ_n^{bound})** in the interfacial region.”

(caption of **Fig. 2, Page 22, lines 618-619**)

Comment 16) Line 109: Typo: water remaining not remained

Response:

We thank the reviewer for pointing this out, and have corrected it as “**remaining**”.

Comment 17) Line 110: ' make the water remained in the interfacial region become bound...' For better readability, please clarify the expression 'bound' here; for example water coordinated to Li^+ cations

Response:

We appreciate the reviewer's constructive suggestion, and have re-added the explanation of “bound” here (*Page 5, lines 119-120*), as:

“make the water **remaining** in the interfacial region become bound **with Li^+** ”.

Comment 18) Line 480 (Figure 3): Please explain in the Figure caption how data shown in subplots b), c), d) was obtained.

Response:

We appreciate the reviewer's constructive suggestion. As for panel b in **Fig. 3**, we analyze the number of H-bonds formed between a water molecule and its surrounded water molecules/cations/anions. Herein, to define the H-bond, a geometrical criterion is used (**Figure R1.16a**),^{42,43} as $r_{\text{HB}} \leq 0.35 \text{ nm}$ and $\alpha_{\text{HB}} \leq 30^\circ$, where r_{HB} is the distance between the possible donor with a hydrogen (H) bonded to it, and acceptor which is not bonded to donor, and α_{HB} is the angle of hydrogen-donor-acceptor.

Figure R1.16 | Geometrical hydrogen bond criterion. **a**, The schematics of geometrical criterion for the hydrogen bond. r_{HB} is the distance between the donor and acceptor, and α_{HB} is the angle of hydrogen-donor-acceptor. **b-d**, The schematics of H-bond by water molecules with water molecules, cations and anions in $[\text{Pyr}_{13}][\text{TFSI}]$.

Specifically, for H-bond between water and water (**Figure R1.16b**), we define the H-bond to be formed between a hydrogen atom of water and an oxygen atom of another water, if the distance between the two oxygen atoms is shorter than 0.35 nm and the angle of hydrogen-oxygen-oxygen is less than 30° . H-bonds between water and ILs can be seen in **Figure R1.16c-d**. For a cation, the H-bond can be defined between water and hydrogen of cation; for an anion, the H-bond is determined between water and an electronegative atoms of anion (acting as acceptor, here are fluorine, nitrogen or oxygen atom).

For panels c and d of **Fig. 3**, we characterize the interaction energies between components A and B based on MD-obtained trajectories. Technologically, the A-B interaction energy, from van der Waals and coulombic interactions, was calculated between component A and component B surrounding A, using a cutoff method (1.2 nm). Such analysis has been used in previous simulation work.^{44,45}

The free water in panel d of **Fig. 3** is determined as the distance between water and Li⁺ larger than 0.25 nm (Details can be seen in response to **Minor Comment 2**).

Briefly summary:

Based on the reviewer's advice, the above description for the interaction energy has been incorporated into the revised manuscript (*Method Section* of the main text, *Page 14, lines 380-383*), and the description of hydrogen bond criterion has been incorporated into the **Supplementary Information**, as **Supplementary Note 9**, with **Supplementary Figure 18** (*Page 21, lines 269-286*).

Specifically, the modification in main text is listed as:

“We characterize the interaction energy between components A and B based on MD-obtained trajectories. Technologically, the A-B interaction energy, coming from van der Waals and coulombic interactions, was calculated between component A and component B surrounding A, using a cutoff method (1.2 nm). Such analysis has been used in previous simulation work.^{65,66}” (*Page 14, lines 380-383*)

New references are added, as:

“65 Chaban, V. V., Andreeva, N. A. & Fileti, E. E. Graphene/ionic liquid ultracapacitors: does ionic size correlate with energy storage performance? *New Journal of Chemistry* 42, 18409-18417 (2018).

66 Zhang, H., Feng, W., Li, C. & Tan, T. Investigation of the Inclusions of Puerarin and Daidzin with β -Cyclodextrin by Molecular Dynamics Simulation. *The Journal of Physical Chemistry B* 114, 4876-4883 (2010).”

The description of hydrogen bond criterion has been incorporated into the **Supplementary Information**, as **Supplementary Note 9**, with **Supplementary Figure 18** (*Page 21, lines 269-286*).

New references for the definition of H-bond are added in **Supplementary Information**, as:

“16 Luzar, A. & Chandler, D. Effect of Environment on Hydrogen Bond Dynamics in Liquid Water. *Physical Review Letters* 76, 928-931 (1996).

17 Vlcek, L. et al. Electric Double Layer at Metal Oxide Surfaces: Static Properties of the Cassiterite–Water Interface. *Langmuir* 23, 4925-4937 (2007).”

Comment 19) Line 134: The text describes that the spectra of water in Figure 3a (bottom panel) corresponds to 'pure water', but the figure label says it is 'spectra of water in humid IL'. Please clarify.

Response:

We thank the reviewer for pointing this out. The spectrum of water in **Fig. 3a** (bottom panel) indeed corresponds to 'pure water'. We have corrected it (caption of **Fig. 3**, *Page 23*, *lines 626-627*), as:

“MD-calculated IR spectra of **O-H bond** in **pure water**, humid [Py_{r13}][TFSI]...”

Comment 20) Line 149: 'The reduced water clusters' or the reduced amount of water clusters, please rephrase for clarity, also the bound water should be clarified as "Li+ bound water".

Response:

We thank the reviewer for pointing this out. As for the first comment, it is indeed the “the reduced amount of water clusters”. As for the second comment, we have rephrased the 'bound water' with a more specific definition as “Li⁺-bound water”.

Brief summary:

We have corrected this sentence (*Page 7*, *lines 176*), as,

“The reduced **amount of** water clusters and the formed **Li⁺**-bound water....”

Comment 21) Line 149-168: 'interaction between water and water' Please use more specific terms when quantifying interaction energy between water molecules, cations, and ILs throughout this paragraph. Do authors mean energy of hydrogen bonding? Please clarify. Same goes for Figure 3 caption (lines 481-489).

Response:

We appreciate the reviewer for this comment. In this work, we quantify the interaction energies between components A and B based on MD-obtained trajectories. Technologically, the A-B interaction energy was calculated between component A and component B surrounding A, using a cutoff method (1.2 nm). This interaction energy is not the energy of H-bond, which comes from van der Waals and coulombic interactions.

Briefly summary:

The above description for the interaction energy has been incorporated into the revised manuscript (*Method Section* of the main text, *Page 14, lines 380-383*), as:

“We characterize the interaction energy between components A and B based on MD-obtained trajectories. Technologically, the A-B interaction energy, coming from van der Waals and coulombic interactions, was calculated between component A and component B surrounding A, using a cutoff method (1.2 nm). Such analysis has been used in previous simulation work.^{65,66}”

New references are added, as:

“65 Chaban, V. V., Andreeva, N. A. & Fileti, E. E. Graphene/ionic liquid ultracapacitors: does ionic size correlate with energy storage performance? *New Journal of Chemistry* 42, 18409-18417 (2018).

66 Zhang, H., Feng, W., Li, C. & Tan, T. Investigation of the Inclusions of Puerarin and Daidzin with β -Cyclodextrin by Molecular Dynamics Simulation. *The Journal of Physical Chemistry B* 114, 4876-4883 (2010).”

Comment 22) Line 163 'water molecules prefer to tangle with Li⁺ ions' please consider replacing 'tangle' with more scientific vocabulary - coordinate etc.

Response:

We appreciate the reviewer for this helpful suggestion, and have modified it, (*Page 7, lines 193-194*), as,

“water molecules prefer to **associate** with Li⁺ ions.”

Comment 23) Line 166 'indicating the reinforced of O-H' What do authors mean by this? Please explain in more detail when O-H gets reinforced and why this suggest lowered water activity?

Response:

We appreciate the reviewer for this comment.

The infrared spectroscopy (IR) with an infrared spectrum is a good approach to monitoring the structural change of molecules. A good one-to-one correspondence between the stretching frequency (ν) and the bond dissociation energy (D) is established as,^{25,46}

$$\nu = 143.3(D - c_1)^{1/2}$$

where c_1 is characteristic of the two bonded atoms. Thus, the bond dissociation energy, also related to the strength of O-H bond, links with the stretching vibrations.^{1-3,25,46} When the peak positions of O-H stretching vibration of water occur at higher wavenumber, more energy is needed to cleave the O-H bond, and the O-H bond is reinforced, suggesting that water has a lower activity.¹⁻³ For instance, in a very recent work published on Nature Materials,³ the infrared spectroscopy was used as a reporter to explore the stability or overpotential of water as, “*The higher wavenumber of light absorbed by the H-O bond of the water indicates that the H-O covalent bond strength is increased, which explains the substantially increased electrochemical stability. This suggests that a higher overpotential is needed to electrochemically decompose water in the presence of the crowding agent PEG*”.

Brief summary:

The above discussion has been incorporated into the revised manuscript (*Page 7, lines 196-198*), as:

“**This indicates that the strength of O-H bond is increased, and then a larger energy is needed to cleave the O-H bond, suggesting that the Li⁺-bound water activity is lowered.**⁴¹⁻⁴³”

New references are added to support this description:

- “41 Badger, R. M. The Relation Between the Energy of a Hydrogen Bond and the Frequencies of the O–H Bands. *The Journal of Chemical Physics* 8, 288-289 (1940).
- 42 Zavitsas, A. A. Quantitative relationship between bond dissociation energies, infrared stretching frequencies, and force constants in polyatomic molecules. *The Journal of Physical Chemistry* 91, 5573-5577 (1987).
- 43 Xie, J., Liang, Z. & Lu, Y.-C. Molecular crowding electrolytes for high-voltage aqueous batteries. *Nature Materials* (2020), doi:10.1038/s41563-020-0667-y.”

Comment 24) Line 495 Figure 4: there is no green shaded region there

Response:

We thank the reviewer for pointing this out. We have corrected “green” as “cyan”.

Comment 25) Line 487 Figure 3c: What is meant by cation here? Li⁺ cation or ionic liquid cation [Pyr13]⁺? Please specify this throughout the text when it might be confusing.

Response:

We thank the reviewer for the suggestion, and we re-edit the expression of cation with a more specific term as [Pyr₁₃]⁺.

Comment 26) Line 230 'makes O-H more stable' What does the stability of O-H mean here? Please clarify

Response:

We appreciate the reviewer for this comment.

Based on the response to **Minor Comment 23**, the strength of O-H bond is related to the stretching vibrations, that is, when the peak positions of stretching vibration of water occur at higher wavenumber, more energy is needed to cleave the O-H bond into radicals, leading to a reinforcement of O-H bond.^{1-3,25,46} In our MD simulations, the peak positions of O-H stretching vibrations of interfacial water in salt-in-humid RTIL are found to shift towards higher wavenumber compared with that of interfacial

water in humid RTIL. Thus, more energy is required to cleave the O-H bond of water, and the O-H bond is reinforced, suggesting that with adding salt, water has a lower activity and thus its electrochemical decomposition needs a higher potential.

Brief summary:

We modified the manuscript (*Page 11, lines 289-291*), as:

“the peak positions of O-H stretching vibrations of interfacial water in salt-in-humid RTIL are shifted towards higher wavenumber, **indicating that O-H bond becomes** more stable than without salt.⁴¹⁻⁴³”

New references are added to support this description:

“41 Badger, R. M. The Relation Between the Energy of a Hydrogen Bond and the Frequencies of the O–H Bands. *The Journal of Chemical Physics* 8, 288-289 (1940).

42 Zavitsas, A. A. Quantitative relationship between bond dissociation energies, infrared stretching frequencies, and force constants in polyatomic molecules. *The Journal of Physical Chemistry* 91, 5573-5577 (1987).

43 Xie, J., Liang, Z. & Lu, Y.-C. Molecular crowding electrolytes for high-voltage aqueous batteries. *Nature Materials* (2020), doi:10.1038/s41563-020-0667-y.”

Comment 27) Line 10: The water remained on the electrode is almost bound - The water (which) remained ´

Response:

We thank the reviewer for the comment, and have re-edited it (*Page 1, line 10*), as,

“The water **remaining** on the electrode is almost bound with Li⁺”

Comment 28) Line 193: find a positive potential well near electrode ´find a positive potential well near (the) electrode ´

Response:

We thank the reviewer for pointing out this. We have corrected as suggested.

Responses to Reviewer #2's Comments

Reviewer #2 (Remarks to the Author):

This manuscript reports a method for expanding the voltage window of humid ionic liquids. The idea, which is very simple, consists in adding a Li ion-based salt. The lithium "traps" the water molecules far from the interface, resulting in a substantial broadening of the electrochemical window. Experiments were performed to validate the idea and simulations allowed to understand the mechanisms at play. This work is very well conducted; it inspires on the series of recent works concerning water-in-salt electrolytes and more generally on studies where water acts as a reactant rather than as a solvent. I am sure it will interest a broad range of scientist and I recommend publication of the manuscript in Nature Communications, provided that the following points are addressed by the authors:

Response:

We really appreciate the reviewer for recognizing the novelty, significance and the potential influence of this work.

Comment 1) I think that the experiments could be consolidated to really prove the importance of the effect. In fact, it is possible that the proposed mechanism affects only the kinetics of the water reduction and/or oxidation, so it would be necessary to show cyclic voltammeteries at slower rate in order to prove that nothing happens.

Response:

We appreciate the reviewer's constructive suggestion.

As suggested, we carried out additional CV measurements for pure [Pyr₁₃][TFSI], humid [Pyr₁₃][TFSI], and salt-in-humid [Pyr₁₃][TFSI], to testify the effect of scan rate on the electrochemical window. We re-prepared the humid RTIL and salt-in-humid IL. The water content for humid [Pyr₁₃][TFSI] is ~4594 ppm (close to 4474 ppm, which is used in main text), and the molar salt-water ratio is 1:1 for salt-in-humid [Pyr₁₃][TFSI]. With scan rates of 5, 10, 50, and 100 mV s⁻¹, cyclic voltammograms of HOPG in three types of electrolytes were shown in **Figure R2.1**. It can be seen that as the scan rate is reduced from 100 to 5 mV s⁻¹, the electrochemical window changes

very little. Therefore, the phenomenon that the electrochemical window of humid RTIL could be enhanced by adding salt is unchanged with varying the scan rate of CV measurements.

Figure R2.1 | Cyclic voltammograms of HOPG at different scan rates. a-c, CV profiles of HOPG for pure [Pyr₁₃][TFSI] (a), humid [Pyr₁₃][TFSI] (b) and salt-in-humid [Pyr₁₃][TFSI] (c) at different scan rates (5, 10, 50 and 100 mV s⁻¹). The water content for humid [Pyr₁₃][TFSI] is ~4594 ppm. The molar salt-water ratio for salt-in-humid [Pyr₁₃][TFSI] electrolyte is 1:1.

Brief summary:

Discussions on the scan rate effect have been incorporated into the revised manuscript (Page 5, lines 130-131). New CVs of HOPG at different scan rates in electrolytes

have been added into **Supplementary Note 7**, with **Supplementary Figure 13** (Pages 15-16, lines 181-194).

Specifically, the modification in main text is listed as:

“This phenomenon is little influenced by changing the scan rate of CV measurements (even down to 5 mV S^{-1} , see **Supplementary Figure 13**).” (Page 5, lines 130-131)

Comment 2) I am quite confused by the discussion of Figure 4 (i.e. page 7). In fact the authors do not say to which system(s) (i.e. concentration) the results correspond to, so it is difficult to understand. In addition, there is a reference to Fig. 3c (line 191) which should in fact be Fig. 4c I think

Response:

We appreciate the reviewer for these useful comments.

To determine the PMF profiles for water, Li^+ or $\text{Li}^+\text{-H}_2\text{O}$, one water molecule, $\text{Li}^+\text{-TFSI}$ pair, or $\text{Li}^+(\text{H}_2\text{O})\text{-TFSI}$ pair is, respectively, added into the ionic liquid system, since the PMF for one particle (which could be an atom, molecule, ion or group) is enough to reveal its free energy distribution and the origin of its density distribution.^{47,48} Such method has been used by many other researchers.^{47,48}

Additionally, the **Fig. 3c** (line 191) is indeed **Fig. 4c**.

Brief summary:

Based on the reviewer’s advice, the description of PMF calculation has been incorporated into the revised manuscript (**Method Section** of the main text, Pages 13-14, lines 374-378), as:

“To determine the PMF profile for water, Li^+ and $\text{Li}^+\text{-H}_2\text{O}$, one water molecule, $\text{Li}^+\text{-TFSI}$ pair, or $\text{Li}^+(\text{H}_2\text{O})\text{-TFSI}$ pair is, respectively, added into the ionic liquid system, since the PMF for one particle (i.e., an atom, molecule, ion or group) is enough to reveal its free energy distribution and the origin of its density

distribution.^{62,64}”

New references related to the PMF have been added, as:

“62 Allen, T. W., Andersen, O. S. & Roux, B. Molecular Dynamics — Potential of Mean Force Calculations as a Tool for Understanding Ion Permeation and Selectivity in Narrow Channels. *Biophysical Chemistry* 124, 251-267 (2006).

64 Yu, Z., Wu, H. & Qiao, R. Electrical Double Layers near Charged Nanorods in Mixture Electrolytes. *The Journal of Physical Chemistry C* 121, 9454-9461 (2017).”

Comment 3) How was the Li-anion binding energy (line 185) calculated?

Response:

We appreciate the reviewer for this comment. The Li-anion binding energy in this work is the intermolecular energy of ion pair, which is calculated based on the interaction of cation and anion within their solvation shell. Such analysis has been used in previous work.⁴⁹

Brief summary:

Description for the calculation of Li-anion binding energy has been added in the revised manuscript (*Page 8, lines 220-223*), as:

“since the intermolecular energy of ion pair, computed by interaction energies between cation and anions within its solvation shell,⁴⁵ reveals that the Li⁺-anion interaction (around -370 kJ mol⁻¹) is much stronger than that of [Pyr₁₃]⁺-[TFSI]⁻ pair (*ca.* -220 kJ mol⁻¹).”

New reference has been added, as:

“45 Kuharski, R. A. & Rossky, P. J. Solvation of Hydrophobic Species in Aqueous Urea Solution: A Molecular Dynamics Study. *Journal of the American Chemical Society* 106, 5794-5800 (1984).”

Comment 4) I think the authors should be a bit more cautious when discussing the interaction energies (page 6). These are effective energies, computed using a classical

force field and for a given structure of the liquid, and not absolute energies. The variations with Li-salt concentrations are interesting, but the relative values between the different terms should not be overinterpreted. In addition the method used to compute these terms should be detailed.

Response:

We thank the reviewer for the constructive comments. Indeed, the interaction energies are effective energies based on the classical force field rather than the absolute ones. We characterize the interaction energies between components A and B based on MD-obtained trajectories. Technologically, the A-B interaction energy, coming from van der Waals and coulombic interactions, was calculated between component A and component B surrounding A, using a cutoff method (1.2 nm). Since the effective interaction energies are calculated with the same process, it is reasonable to compare these values between water and other components (including water, [Pyr₁₃]⁺, [TFSI]⁻ and Li⁺) and then determine the strength of the interaction between water and other components. Such method for analyzing interaction energy has been used in previous simulation work.^{44,45}

Brief summary:

With the reviewer's useful suggestion, the above description for the interaction energy has been incorporated into the revised manuscript (*Method Section* of the main text, *Page 14, lines 380-383*), as:

“We characterize the interaction energies between components A and B based on MD-obtained trajectories. Technologically, the A-B interaction energy, from van der Waals and coulombic interactions, was calculated between component A and component B surrounding A, using a cutoff method (1.2 nm). Such analysis has been used in previous simulation work.^{65,66}”

New references are added, as:

“65 Chaban, V. V., Andreeva, N. A. & Fileti, E. E. Graphene/ionic liquid ultracapacitors: does ionic size correlate with energy storage performance? *New Journal of Chemistry* 42, 18409-18417 (2018).

66 Zhang, H., Feng, W., Li, C. & Tan, T. Investigation of the Inclusions of Puerarin and Daidzin with β -Cyclodextrin by Molecular Dynamics Simulation. *The Journal of Physical Chemistry B* 114, 4876-4883 (2010).”

Comment 5) There are quite a lot of English mistakes that should be corrected:

-line 74: "distributions"

-line 84: "to closely stay" does not seem right

-line 92: "negative"

-line 109: "remain"

-line 126: I do not understand "deployed" in this context

-line 173: "by using the"

-line 183: "could be attributed"

-line 205: Sentence should be rephrased ("attributed to that" does not seem right)

-line 270: do the authors really mean "adapted"? Or "adopted"?

Response:

We appreciate the reviewer for carefully reading. We have corrected them in the manuscript (*Page 3, lines 77-78; Page 4, lines 88-89; Page 4, line 98; Page 5, line 119; Page 6, line 140; Page 8, line 205; Page 8, line 219; Page 9, line 244; Page 13, line 346*), as:

“Panels c and f in **Fig. 1** exhibit water **distributions** with...” (*Page 3, lines 77-78*)

“found to stay **closely** with the added Li^+ ions” (*Page 4, lines 88-89*)

“As the voltage becomes more **negative**,” (*Page 4, line 98*)

“the water **remaining** in the interfacial” (*Page 5, line 119*)

“which was **used** as the infrared spectroscopy (IR) reporter ...” (*Page 6, line 140*)

“**by using the** umbrella sampling method” (*Page 8, line 205*)

“**may be** attributed to Li^+ ions...” (*Page 8, line 219*)

“expansion of electrochemical window could be **ascribed** to...” (*Page 9, line 244*)

“SPC/E model was **adopted** for water molecules” (*Page 13, line 346*)

References

- 1 Yamada, Y. *et al.* Hydrate-Melt Electrolytes for High-Energy-Density Aqueous Batteries. *Nature Energy* **1**, 16129-16137 (2016).
- 2 Dubouis, N. *et al.* The Role of the Hydrogen Evolution Reaction in the Solid-Electrolyte Interphase Formation Mechanism for “Water-in-Salt” Electrolytes. *Energy & Environmental Science* **11**, 3491-3499 (2018).
- 3 Xie, J., Liang, Z. & Lu, Y.-C. Molecular Crowding Electrolytes for High-Voltage Aqueous Batteries. *Nature Materials* (2020), doi.org/10.1038/s41563-020-0667-y.
- 4 Vatamanu, J. & Borodin, O. Ramifications of Water-in-Salt Interfacial Structure at Charged Electrodes for Electrolyte Electrochemical Stability. *Journal of Physical Chemistry Letters* **8**, 4362-4367 (2017).
- 5 Zheng, J. *et al.* Understanding Thermodynamic and Kinetic Contributions in Expanding the Stability Window of Aqueous Electrolytes. *Chem* **4**, 2872-2882 (2018).
- 6 Kumar, S., Rosenberg, J. M., Bouzida, D., Swendsen, R. H. & Kollman, P. A. The Weighted Histogram Analysis Method for Free-Energy Calculations on Biomolecules. I. The Method. *Journal of Computational Chemistry* **13**, 1011-1021 (1992).
- 7 Kästner, J. Umbrella Sampling. *WIREs Computational Molecular Science* **1**, 932-942 (2011).
- 8 Grimme, S., Antony, J., Ehrlich, S. & Krieg, H. A Consistent and Accurate Ab Initio Parametrization of Density Functional Dispersion Correction (DFT-D) for the 94 Elements H-Pu. *Journal of Chemical Physics* **132**, 154104 (2010).
- 9 Kresse, G. & Furthmüller, J. Efficiency of Ab-Initio Total Energy Calculations for Metals and Semiconductors Using a Plane-Wave Basis Set. *Computational Materials Science* **6**, 15-50 (1996).
- 10 Perdew, J. P., Burke, K. & Ernzerhof, M. Generalized Gradient Approximation Made Simple. *Physical Review Letters* **77**, 3865-3868 (1996).
- 11 Blöchl, P. E. Projector Augmented-Wave Method. *Physical Review B* **50**, 17953-17979 (1994).
- 12 Voloshina, E., Usvyat, D., Schutz, M., Dedkov, Y. & Paulus, B. On the Physisorption of Water on Graphene: A CCSD(T) Study. *Physical Chemistry Chemical Physics* **13**, 12041-12047 (2011).
- 13 Manz, T. A. & Sholl, D. S. Chemically Meaningful Atomic Charges That Reproduce the Electrostatic Potential in Periodic and Nonperiodic Materials. *Journal of Chemical Theory and Computation* **6**, 2455-2468 (2010).
- 14 Manz, T. A. & Sholl, D. S. Improved Atoms-in-Molecule Charge Partitioning

- Functional for Simultaneously Reproducing the Electrostatic Potential and Chemical States in Periodic and Nonperiodic Materials. *Journal of Chemical Theory and Computation* **8**, 2844-2867 (2012).
- 15 Manz, T. A. & Limas, N. G. Introducing DDEC6 Atomic Population Analysis: Part 1. Charge Partitioning Theory and Methodology. *RSC Advances* **6**, 47771-47801 (2016).
 - 16 Henderson, W. A. & Passerini, S. Phase Behavior of Ionic Liquid–LiX Mixtures: Pyrrolidinium Cations and TFSI- Anions. *Chemistry of Materials* **16**, 2881-2885 (2004).
 - 17 Velasco-Velez, J.-J. *et al.* The Structure of Interfacial Water on Gold Electrodes Studied by X-Ray Absorption Spectroscopy. *Science* **346**, 831 (2014).
 - 18 Li, C.-Y. *et al.* In Situ Probing Electrified Interfacial Water Structures at Atomically Flat Surfaces. *Nature Materials* **18**, 697-701 (2019).
 - 19 Suo, L. *et al.* “Water-in-Salt” Electrolyte Makes Aqueous Sodium-Ion Battery Safe, Green, and Long-Lasting. *Advanced Energy Materials* **7**, 1701189-1701198 (2017).
 - 20 Yaghini, N., Pitawala, J., Matic, A. & Martinelli, A. Effect of Water on the Local Structure and Phase Behavior of Imidazolium-Based Protic Ionic Liquids. *Journal of Physical Chemistry B* **119**, 1611-1622 (2015).
 - 21 Cheng, H.-W. *et al.* Characterizing the Influence of Water on Charging and Layering at Electrified Ionic-Liquid/Solid Interfaces. *Advanced Materials Interfaces* **2**, 1500159 (2015).
 - 22 Bi, S. *et al.* Minimizing the Electrosorption of Water from Humid Ionic Liquids on Electrodes. *Nature Communications* **9**, 5222-5230 (2018).
 - 23 Stettner, T., Gehrke, S., Ray, P., Kirchner, B. & Balducci, A. Water in Protic Ionic Liquids: Properties and Use of a New Class of Electrolytes for Energy-Storage Devices. *ChemSusChem* **12**, 3827-3836 (2019).
 - 24 Badger, R. M. The Relation between the Energy of a Hydrogen Bond and the Frequencies of the O–H Bands. *The Journal of Chemical Physics* **8**, 288-289 (1940).
 - 25 Zavitsas, A. A. Quantitative Relationship between Bond Dissociation Energies, Infrared Stretching Frequencies, and Force Constants in Polyatomic Molecules. *The Journal of Physical Chemistry* **91**, 5573-5577 (1987).
 - 26 Zhao, S. *et al.* Quantitative Determination on Ionic-Liquid-Gating Control of Interfacial Magnetism. *Advanced Materials* **29**, 1606478 (2017).
 - 27 Fajardo, O. Y., Bresme, F., Kornyshev, A. A. & Urbakh, M. Water in Ionic Liquid Lubricants: Friend and Foe. *ACS Nano* **11**, 6825-6831 (2017).
 - 28 Millefiorini, S., Tkaczyk, A. H., Sedev, R., Efthimiadis, J. & Ralston, J.

- Electrowetting of Ionic Liquids. *Journal of the American Chemical Society* **128**, 3098-3101 (2006).
- 29 Suo, L. *et al.* “Water-in-Salt” Electrolyte Enables High-Voltage Aqueous Lithium-Ion Chemistries. *Science* **350**, 938-943 (2015).
- 30 Dou, Q. *et al.* Safe and High-Rate Supercapacitors Based on an “Acetonitrile/Water in Salt” Hybrid Electrolyte. *Energy & Environmental Science* **11**, 3212-3219 (2018).
- 31 Borodin, O., Self, J., Persson, K. A., Wang, C. & Xu, K. Uncharted Waters: Super-Concentrated Electrolytes. *Joule* **4**, 69-100 (2020).
- 32 Lukatskaya, M. R. *et al.* Concentrated Mixed Cation Acetate “Water-in-Salt” Solutions as Green and Low-Cost High Voltage Electrolytes for Aqueous Batteries. *Energy & Environmental Science* **11**, 2876-2883 (2018).
- 33 Dou, Q. *et al.* A Sodium Perchlorate-Based Hybrid Electrolyte with High Salt-to-Water Molar Ratio for Safe 2.5 V Carbon-Based Supercapacitor. *Energy Storage Materials* **23**, 603-609 (2019).
- 34 Zhao, C., Bond, A. M. & Lu, X. Determination of Water in Room Temperature Ionic Liquids by Cathodic Stripping Voltammetry at a Gold Electrode. *Analytical Chemistry* **84**, 2784-2791 (2012).
- 35 Costa, R., Pereira, C. M. & Silva, A. F. Dicationic Ionic Liquid: Insight in the Electrical Double Layer Structure at Mercury, Glassy Carbon and Gold Surfaces. *Electrochimica Acta* **116**, 306-313 (2014).
- 36 Zhang, Q. *et al.* Electrochemical Impedance Spectroscopy on the Capacitance of Ionic Liquid–Acetonitrile Electrolytes. *Electrochimica Acta* **270**, 352-362 (2018).
- 37 Gnahn, M., Pajkossy, T. & Kolb, D. M. The Interface between Au(111) and an Ionic Liquid. *Electrochimica Acta* **55**, 6212-6217 (2010).
- 38 McEldrew, M., Goodwin, Z. A. H., Kornyshev, A. A. & Bazant, M. Z. Theory of the Double Layer in Water-in-Salt Electrolytes. *Journal of Physical Chemistry Letters* **9**, 5840-5846 (2018).
- 39 Black, J. M. *et al.* Bias-Dependent Molecular-Level Structure of Electrical Double Layer in Ionic Liquid on Graphite. *Nano Letters* **13**, 5954-5960 (2013).
- 40 Haskins, J. B., Wu, J. J. & Lawson, J. W. Computational and Experimental Study of Li-Doped Ionic Liquids at Electrified Interfaces. *The Journal of Physical Chemistry C* **120**, 11993-12011 (2016).
- 41 Vatamanu, J., Borodin, O. & Smith, G. D. Molecular Insights into the Potential and Temperature Dependences of the Differential Capacitance of a Room-Temperature Ionic Liquid at Graphite Electrodes. *Journal of the American Chemical Society* **132**, 14825-14833 (2010).
- 42 Luzar, A. & Chandler, D. Effect of Environment on Hydrogen Bond Dynamics in

- Liquid Water. *Physical Review Letters* **76**, 928-931 (1996).
- 43 Vlcek, L. *et al.* Electric Double Layer at Metal Oxide Surfaces: Static Properties of the Cassiterite–Water Interface. *Langmuir* **23**, 4925-4937 (2007).
- 44 Chaban, V. V., Andreeva, N. A. & Fileti, E. E. Graphene/Ionic Liquid Ultracapacitors: Does Ionic Size Correlate with Energy Storage Performance? *New Journal of Chemistry* **42**, 18409-18417 (2018).
- 45 Zhang, H., Feng, W., Li, C. & Tan, T. Investigation of the Inclusions of Puerarin and Daidzin with B-Cyclodextrin by Molecular Dynamics Simulation. *The Journal of Physical Chemistry B* **114**, 4876-4883 (2010).
- 46 Bernstein, H. J. The Average Xh Stretching Frequency as a Measure of Xh Bond Properties. *Spectrochimica Acta* **18**, 161-170 (1962).
- 47 Yu, Z., Wu, H. & Qiao, R. Electrical Double Layers near Charged Nanorods in Mixture Electrolytes. *The Journal of Physical Chemistry C* **121**, 9454-9461 (2017).
- 48 Allen, T. W., Andersen, O. S. & Roux, B. Molecular Dynamics — Potential of Mean Force Calculations as a Tool for Understanding Ion Permeation and Selectivity in Narrow Channels. *Biophysical Chemistry* **124**, 251-267 (2006).
- 49 Kuharski, R. A. & Rossky, P. J. Solvation of Hydrophobic Species in Aqueous Urea Solution: A Molecular Dynamics Study. *Journal of the American Chemical Society* **106**, 5794-5800 (1984).

REVIEWERS' COMMENTS

Reviewer #1 (Remarks to the Author):

The authors invested a lot of time and care in answering my questions. The answers and updates to the manuscript are satisfying and I feel that this work is ready for publication in Nature Comm. I am sure this work will attract a lot of interest.

Reviewer #2 (Remarks to the Author):

I think that the authors did good efforts to strengthen the manuscript and answer the various points raised by both referees. I recommend publication in Nature Communications, but I still have a few remaining (minor) comments:

1/ I am not completely convinced by the DFT calculations and the conclusions drawn from them. Firstly, I do not think that the HOMO is the relevant quantity to look at in this context. When discussing a reaction it is important to determine the free energy variation when passing from the reactants to the products, which would require much more involved calculations such as the ones done for TiO₂/water interfaces by Sprik, Cheng and co-workers. Similarly, computing simple energies may give a hint but I do not think it is enough. I think it would be good to mention this limitation in the manuscript.

2/ When discussing the reactivity of water (either in the Li-free or in the Li-loaded RTIL), the authors are a bit vague. Recent works have shown that isolated water molecules have a different reactivity than aqueous clusters. In the latter the H-bond network allows to separate efficiently the water products. The case of Li-bounded molecule is an intermediate: The reactivity of the water molecules depends on both species (Li and H₂O) concentrations in a non-trivial way, both because of the strength of the Li-H₂O reaction (as discussed here) and of the ability to form water "domains". This issue should also be clarified.

3/ Although the English was improved w.r.t the first version, some parts of the text remain a bit difficult to read (maybe this point can be addressed by the editor).

We've made point-to-point response to each comment. Specifically, reviewers' comments are copied in blue, and each comment is followed by our response in black. The manuscript has been revised, accordingly, in red. After each response to comment, a brief summary is provided of what has been changed or added and where they are positioned in the revised manuscript and corresponding Supplementary Information. Furthermore, legends of the figures are modified to contain definitions of abbreviations shown in the figures.

Reviewer #1 (Remarks to the Author):

The authors invested a lot of time and care in answering my questions. The answers and updates to the manuscript are satisfying and I feel that this work is ready for publication in Nature Comm. I am sure this work will attract a lot of interest.

Response: We thank this reviewer very much for her/his positive comment and recommendation to Nature Communications.

Reviewer #2 (Remarks to the Author):

I think that the authors did good efforts to strengthen the manuscript and answer the various points raised by both referees. I recommend publication in Nature Communications, but I still have a few remaining (minor) comments:

Response: We appreciate that the reviewer re-evaluated our work and recommended our work for publication in Nature Communications.

1/ I am not completely convinced by the DFT calculations and the conclusions drawn from them. Firstly, I do not think that the HOMO is the relevant quantity to look at in this context. When discussing a reaction it is important to determine the free energy variation when passing from the reactants to the products, which would require much more involved calculations such as the ones done for TiO₂/water interfaces by Sprik, Cheng and co-workers. Similarly, computing simple energies may give a hint but I do not think it is enough. I think it would be good to mention this limitation in the manuscript.

Response: We appreciate the reviewer for these thoughtful comments. We agree with the reviewer that it would be better to calculate the free energy variation between the reactants and products when discussing a reaction, such as the studies by Sprik *et al.*^{1,2} and Borodin *et al.*³ who quantitatively analyzed chemical reactions of the proton transfer near TiO₂/H₂O interface^{1,2} and the solvent oxidation³, including the analysis of HOMO level.

In fact, the free energy change under positive polarization is the oxidation energy. Koch group⁴ and Ue group⁵ have established qualitative correlations between the oxidation energy and the HOMO level: the lower HOMO leads to higher oxidation energy, indicating more energy is required for the oxidation reaction to occur. Thus, the trend in the calculated HOMO level could be compatible with that of oxidation energy, and such analysis has been adopted in qualitatively determining the oxidative stability of ILs⁵⁻⁷ and organic molecule by complexation with Li⁺ cation.⁸ Therefore, the calculated HOMO level could be considered as a qualitative approach to estimating the oxidation stability of water in humid IL and in salt-in-humid IL electrolytes.

Brief Summary:

Following the reviewer's comments, the above discussion has been incorporated into the revised manuscript (*Page 5, lines 112-119*), as:

“Thus, the lowered HOMO level by adding salt would help to enhance the oxidation stability of water at positive electrode. It is worth noting that to quantitatively evaluate such oxidation stability, the oxidation potential could be analyzed from the free energy variation between the reactants and products under positive polarization,³⁵⁻³⁷ which has been correlated with the HOMO level: the lower HOMO leads to higher oxidation energy, indicating more energy is required for the oxidation reaction to occur.^{38,39} Therefore, the calculated HOMO level could be considered as a qualitative approach to estimating the oxidation stability of water in electrolyte. Moreover, ...”

New references have been added as:

“35 Cheng, J. & Sprik, M. Aligning Electronic Energy Levels at the TiO₂/H₂O Interface. *Physical Review B* **82**, 081406 (2010).

36 Cheng, J. & Sprik, M. Alignment of Electronic Energy Levels at Electrochemical Interfaces. *Physical Chemistry Chemical Physics* **14**, 11245-11267 (2012).

37 Borodin, O. *et al.* Modeling Insight into Battery Electrolyte Electrochemical Stability and Interfacial Structure. *Accounts of Chemical Research* **50**, 2886-2894 (2017).

38 Koch, V. R., Dominey, L. A., Nanjundiah, C. & Ondrechen, M. J. The Intrinsic Anodic Stability of Several Anions Comprising Solvent-Free Ionic Liquids. *Journal of the Electrochemical Society* **143**, 798-803 (1996).

39 Ue, M., Murakami, A. & Nakamura, S. Anodic Stability of Several Anions Examined by Ab Initio Molecular Orbital and Density Functional Theories. *Journal of the Electrochemical Society* **149**, A1572 (2002).”

2/ When discussing the reactivity of water (either in the Li-free or in the Li-loaded RTIL), the authors are a bit vague. Recent works have shown that isolated water molecules have a different reactivity than aqueous clusters. In the latter the H-bond network allows to separate efficiently the water products.

The case of Li-bounded molecule is an intermediate: The reactivity of the water molecules depends on both species (Li and H₂O) concentrations in a non-trivial way, both because of the strength of the Li-H₂O reaction (as discussed here) and of the ability to form water "domains". This issue should also be clarified.

Response: We thank the reviewer for the thoughtful comments. We agree that isolated water molecules have different reactivity from aqueous clusters, either in the Li-free or in Li-loaded IL system. In a very recent work, Dubouis *et al.*⁹ demonstrated that the water reactivity of system

can be controlled by tuning the interaction of water-water and water-salt, that is, the reactivity for water is the highest in water-rich domains (or aqueous clusters) owing to the autoprotolysis mechanism,^{10,11} since the water electrolysis products (proton and hydroxide ions) can be separated efficiently through the H-bond network, following a Grotthuss diffusion mechanism.⁹ Our simulation results show that the water molecules exist in the form of water clusters (**Figure R1a**), and the H-bond network can be formed in humid IL electrolyte (**Figure R2**). However, with adding Li-salt, the H-bond network nearly disappeared (**Figure R2**), leading to an inefficient separation of the water electrolysis products and then lowered reactivity.

Figure R1 | Water cluster in different electrolytes. **a**, The cluster distribution of water in humid [Py_{r13}][TFSI]. **b-c**, The cluster distribution of water and its association with Li⁺ when the molar salt-water ratios are 1:1 (**b**) and 2:1 (**c**). This figure is copied from **Supplementary Figure 19**.

Figure R2 | H-bonds of water in different electrolytes. H-bonds between water molecules (left axis) and between water molecules and IL ions (right axis). This figure is copied from **Fig. 3b** in the main text.

It is true that the reactivity of the water molecules depends on both species (Li⁺ and H₂O) concentration. Water molecules could form water cluster in humid IL (**Figure R1a**). However, rather than forming water “domains”, water molecules in salt-in-humid IL electrolytes tend to associate with Li⁺ ions and become “bound” water, leading to the remarkably decreased water clusters (**Figure R1b-c**). Furthermore, the strength of O-H bond in bound water is increased owing to the association with Li⁺ and thus more energy is needed to split the O-H bond.¹² Therefore, the reactivity of Li⁺-bound water is decreased.

Briefly, compared with humid IL, the nearly disappeared H-bond network and largely reduced amount of water clusters could decrease the activity of water in salt-in-humid IL electrolytes.

Brief Summary:

Following the reviewer's comments, the above discussion has been incorporated into the revised manuscript (*Page 7, lines 182-187*), as:

“Furthermore, the H-bond network in water cluster, following a Grotthuss diffusion mechanism, helps to efficiently separate water electrolysis products,⁴⁹⁻⁵¹ thus enhancing the water activity.⁵¹ Hence, compared with humid IL, the nearly disappeared H-bond network (**Fig. 3b**) and largely reduced amount of water clusters (**Supplementary Figure 19b-c**) could decrease the activity of water in salt-in-humid IL electrolytes.”

New references have been added as:

“49 Marx, D., Tuckerman, M. E., Hutter, J. & Parrinello, M. The Nature of the Hydrated Excess Proton in Water. *Nature* **397**, 601-604 (1999).

50 Tuckerman, M., Marx, D. & Parrinello, M. The Nature and Transport Mechanism of Hydrated Hydroxide Ions in Aqueous Solution. *Nature* **417**, 925-929 (2002).

51 Dubouis, N. *et al.* Tuning Water Reduction through Controlled Nanoconfinement within an Organic Liquid Matrix. *Nature Catalysis* (2020), DOI: 10.1038/s41929-020-0482-5.”

3/ Although the English was improved w.r.t the first version, some parts of the text remain a bit difficult to read (maybe this point can be addressed by the editor).

Response: We appreciate the reviewer for carefully reading. The manuscript has been revised for better reading, and some key changes are listed as:

- 1) “However, it is still an unaddressed issue that, for hydrophobic ILs that have been widely used as electrolytes...” (*Page 2, line 40*)
- 2) “This is ascribed to the notable decrease of water activity, since the strong interaction between water and Li⁺ ion leads to a large shrinkage of “free” water (i.e., water is not bound with Li⁺).” (*Page 2, lines 46-48*)

- 3) "... alternating layers extending up to a few nanometers from the electrode surfaces"
(Page 3, line 73)
- 4) "we first explore the underlying origin of the Li⁺-bound water and reinforcement of O-H bond." (Page 6, line 145)

References

- 1 Cheng, J. & Sprik, M. Aligning Electronic Energy Levels at the TiO₂/H₂O Interface. *Physical Review B* **82**, 081406 (2010).
- 2 Cheng, J. & Sprik, M. Alignment of Electronic Energy Levels at Electrochemical Interfaces. *Physical Chemistry Chemical Physics* **14**, 11245-11267 (2012).
- 3 Borodin, O. *et al.* Modeling Insight into Battery Electrolyte Electrochemical Stability and Interfacial Structure. *Accounts of Chemical Research* **50**, 2886-2894 (2017).
- 4 Koch, V. R., Dominey, L. A., Nanjundiah, C. & Ondrechen, M. J. The Intrinsic Anodic Stability of Several Anions Comprising Solvent - Free Ionic Liquids. *Journal of the Electrochemical Society* **143**, 798-803 (1996).
- 5 Ue, M., Murakami, A. & Nakamura, S. Anodic Stability of Several Anions Examined by Ab Initio Molecular Orbital and Density Functional Theories. *Journal of the Electrochemical Society* **149**, A1572 (2002).
- 6 Maeshima, H., Moriwake, H., Kuwabara, A. & Fisher, C. A. J. Quantitative Evaluation of Electrochemical Potential Windows of Electrolytes for Electric Double-Layer Capacitors Using Ab Initio Calculations. *Journal of the Electrochemical Society* **157**, A696-A701 (2010).
- 7 Ong, S. P. & Ceder, G. Investigation of the Effect of Functional Group Substitutions on the Gas-Phase Electron Affinities and Ionization Energies of Room-Temperature Ionic Liquids Ions Using Density Functional Theory. *Electrochimica Acta* **55**, 3804-3811 (2010).
- 8 Yoshida, K. *et al.* Oxidative-Stability Enhancement and Charge Transport Mechanism in Glyme-Lithium Salt Equimolar Complexes. *Journal of the American Chemical Society* **133**, 13121-13129 (2011).
- 9 Dubouis, N. *et al.* Tuning Water Reduction through Controlled Nanoconfinement within an Organic Liquid Matrix. *Nature Catalysis* (2020), DOI: 10.1038/s41929-020-0482-5.
- 10 Marx, D., Tuckerman, M. E., Hutter, J. & Parrinello, M. The Nature of the Hydrated Excess Proton in Water. *Nature* **397**, 601-604 (1999).
- 11 Tuckerman, M., Marx, D. & Parrinello, M. The Nature and Transport Mechanism of Hydrated Hydroxide Ions in Aqueous Solution. *Nature* **417**, 925-929 (2002).
- 12 Xie, J., Liang, Z. & Lu, Y.-C. Molecular Crowding Electrolytes for High-Voltage Aqueous Batteries. *Nature Materials* **19**, 1006-1011 (2020).